# Chemical heterogeneities reveal early rapid cooling of Apollo Troctolite 76535

William S. Nelson [1✉], Julia E. Hammer [1], Thomas Shea [1], Eric Hellebrand[2] & G. Jeffrey Taylor [1]

The evolution of the lunar interior is constrained by samples of the magnesian suite of rocks returned by the Apollo missions. Reconciling the paradoxical geochemical features of this suite constitutes a feasibility test of lunar differentiation models. Here we present the results of a microanalytical examination of the archetypal specimen, troctolite 76535, previously thought to have cooled slowly from a large magma body. We report a degree of intra-crystalline compositional heterogeneity (phosphorus in olivine and sodium in plagioclase) fundamentally inconsistent with prolonged residence at high temperature. Diffusion chronometry shows these heterogeneities could not have survived magmatic temperatures for >~20 My, i.e., far less than the previous estimated cooling duration of >100 My. Quantitative modeling provides a constraint on the thermal history of the lower lunar crust, and the textural evidence of dissolution and reprecipitation in olivine grains supports reactive melt infiltration as the mechanism by which the magnesian suite formed.

[1] Department of Earth Sciences, School of Ocean and Earth Science and Technology (SOEST), University of Hawaii at Mānoa, Honolulu, HI, USA. [2] Department of Earth Sciences, Utrecht University, Utrecht, The Netherlands. ✉email: wnelson@hawaii.edu

Lunar evolution has been defined by the samples returned by the Apollo missions, culminating in the generally accepted lunar magma ocean (LMO) model. This model postulates that the Moon was once largely molten, and the lithologies now present on the Moon are produced from geochemical reservoirs defined by crystallization products of this melt. However, the magnesian suite (Mg-suite) was found to possess characteristics that prompted amendments to the basic LMO model. This suite contains the combination of some of the most anorthitic plagioclase ($An_{98-84}$) and the most Mg-rich mafic silicates (Mg# 95–60) ever found in lunar samples[1], suggesting crystallization from a parent melt that is highly primitive (unmodified by fractional crystallization). However, these rocks are also enriched in potassium, rare earth elements, and phosphorus (KREEP)[1], which indicate a highly evolved (influenced by fractional crystallization) parent melt. Any plausible model of lunar magmatic evolution must explain these paradoxical chemical features.

Various hypotheses for Mg-suite petrogenesis have been proposed (c.f., ref. [1]). Early studies postulated its direct formation during crystallization of the LMO[2–4]. The resulting models succeeded in generating primitive mineral compositions but required late-stage metasomatism to introduce the KREEP signature. Alternatively, Longhi[5] suggested the Mg-suite was formed by high-pressure partial melting of a bulk-Moon source composition, followed by assimilation of anorthite and a highly evolved melt. Prissel and Gross[6] modified this theory, eliminating the requirement of anorthite crust assimilation, and demonstrating that partial melting at high pressure after cumulate overturn could produce a suitable Mg-suite parent melt. This liquid could then have been extracted from its source, as has been proposed for the younger picritic glasses[7]. Warren and Wasson[8] postulated the existence of a layer of late-stage fluids called "ur-KREEP", which could have been intersected by the Mg-suite parent melts, with the two melts mixing and imparting a KREEP signature to the Mg-suite parent[9,10]. Ringwood and Kesson[11] suggested that overturn of magma ocean cumulates brought solid Mg-rich cumulates near the lunar surface. Elardo et al.[12,13] demonstrated that mixing these cumulates with ur-KREEP would lower the liquidus of the system, and radiogenic heat from the decay of potassium and thorium would have also helped produce the Mg-suite parent melt. Another possibility is that cooling and crystallization of an impact melt sheet generated during an impact event[14,15] caused direct mixing of anorthite-rich crust, ur-KREEP, and mantle material. The variety of mechanisms still being debated after 45 years demonstrates the need for additional constraints on the controlling variables of Mg-suite formation: the timing and process of melt production, the style and geometry of magma emplacement, and the thermal evolution of the lower crust. Chemical zoning within mineral grains can be leveraged to determine the maximum length of time the grains could have remained at magmatic temperatures without being compositionally homogenized by element diffusion[16]. This study constrains the highest temperature portion of the Mg-suite cooling path by evaluating chemical heterogeneities imparted in olivine and plagioclase crystals in troctolite 76535 during their growth from a silicate liquid.

Apollo 17 astronauts collected troctolite 76535 at geology station six[17]. It is a coarse-grained phaneritic rock composed of plagioclase (60%), olivine (35%), and orthopyroxene (5%)[18]. This sample is the most pristine (displaying the least evidence for impact modification) in the Mg-suite, and thus has become one of the most scientifically valuable samples of the Moon[1]. Notably, 76535 has been used to quantify the amount of water in the primitive Moon[19], the evolution and cessation of the lunar magnetic dynamo[20], and the effect of shock on radiometric ages[21].

Our current understanding of this sample's cooling history presents a paradox. Radiometric, petrographic, and petrologic studies of the sample indicate slow subsolidus cooling rates and long cooling times. Age determinations from independent isotopic systems suggest subsolidus cooling rates around 3.9 degrees per million years[22]. However, radiogenic age determinations are not suitable for determining the higher temperature (magmatic) cooling history of this sample. If the cooling path defined by a three-point line connecting paired radiometric closure temperatures and ages[22] is extrapolated to higher temperatures, the sample would have begun crystallizing around $4,421 \pm 11$ Ma, before the estimated $4383 \pm 17$ Ma formation of sample 60025, the best-constrained member of the ferroan anorthosite suite (FAS)[22,23]. As the FAS is postulated to represent the flotation anorthosite crust, this line of reasoning poses a conundrum: The Mg-suite would have begun crystallizing before formation of the crust into which it was emplaced[22].

In this study of troctolite 76535, we present new evidence of subtle chemical heterogeneities in plagioclase and olivine crystals, which we interpret to represent arrested diffusive homogenization. Applying the technique of diffusion chronometry, we conduct extensive numerical modeling to find the maximum amount of time this sample remained above the closure temperature for Na–Ca interdiffusion in plagioclase and P diffusion in olivine. Our results constrain the magmatic thermal history of this sample to <~20 My. By extension to the entire suite, these results challenge the notion that the Mg-suite lithologies represent layers of large, slowly cooled igneous intrusions. Rather, the faster magmatic cooling duration, along with new evidence of crystal dissolution and reprecipitation, suggests the Mg-suite formed by a process involving reactive melt infiltration.

## Results

**Chemical heterogeneities in 76535.** Previous petrologic studies of 76535 report chemical homogeneity of its major phases[1,24], a finding that has gone unchallenged since the 1970s, likely because it is consistent with the sample's extensively annealed texture. Intragrain compositional variations can now be detected at much finer spectral and spatial resolution. Electron probe microanalysis (EPMA) X-ray mapping and profiling on slides of 76535 reveal compositional heterogeneities (Figs. 1 and 2) in nearly every crystal that we mapped. Concentric sodium (Na) zoning in plagioclase is very subtle (given the range in Na) but resolvable. Phosphorus (P) heterogeneities in olivine are lamellar and sharp, similar in appearance to those in terrestrial samples[25–28], as well as lunar basalts[29] and soils[30]. The existence of these heterogeneities implies rapid initial olivine growth[26], which is most readily induced by relatively rapid cooling[31] at magmatic temperatures, and preservation of initial growth zoning down to the closure temperature of P diffusion.

All olivine crystals we analyzed possess regions of P-enriched lamellae alternating with relatively P-poor zones, and regions that are homogeneous and P-poor throughout (Fig. 1). There are no differences in the concentrations of Ca, Fe, Mg, Al, and Cr (Supplementary Fig. 6), or crystallographic orientation (Supplementary Fig. 8) between the two regions. The boundaries between the two types of regions are sharp and cuspate, a cross-cutting relationship that indicates dissolution of the P-enriched olivine and precipitation (i.e., overgrowth) of P-poor olivine. In detail, the gradient across the dissolution front is significantly sharper than across the P-rich lamellar heterogeneities. If the lamellae were sharper when the P enrichments formed[26,28] (during crystal growth), the steeper concentration gradient across the cuspate boundaries suggests that dissolution occurred after most of the diffusive relaxation in the P-enriched regions had occurred.

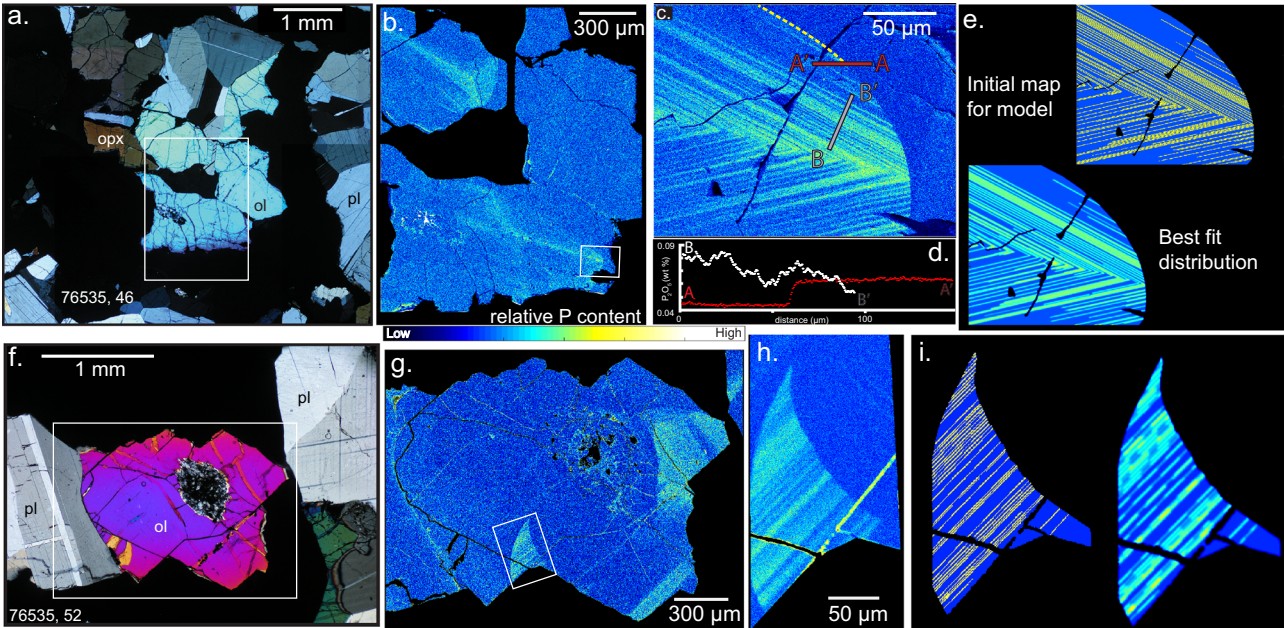

**Fig. 1 Olivine crystals.** Cross-polarized light (XPL) images (**a**, **f**), P kα intensity maps (**b**, **c**, **g**, **h**), profiles (**d**) (red: averaged along truncation surface, white: along line), and diffusion models' initial conditions and final results (**e**, **i**).

**Fig. 2 Plagioclase crystals.** XPL context (**a**, **e**), Na kα X-ray intensity maps (**b**, **f**), profiles (black dots, **d**, **h**), and initial/final diffusion models (**c**, **g** and red lines in **d**, **h**).

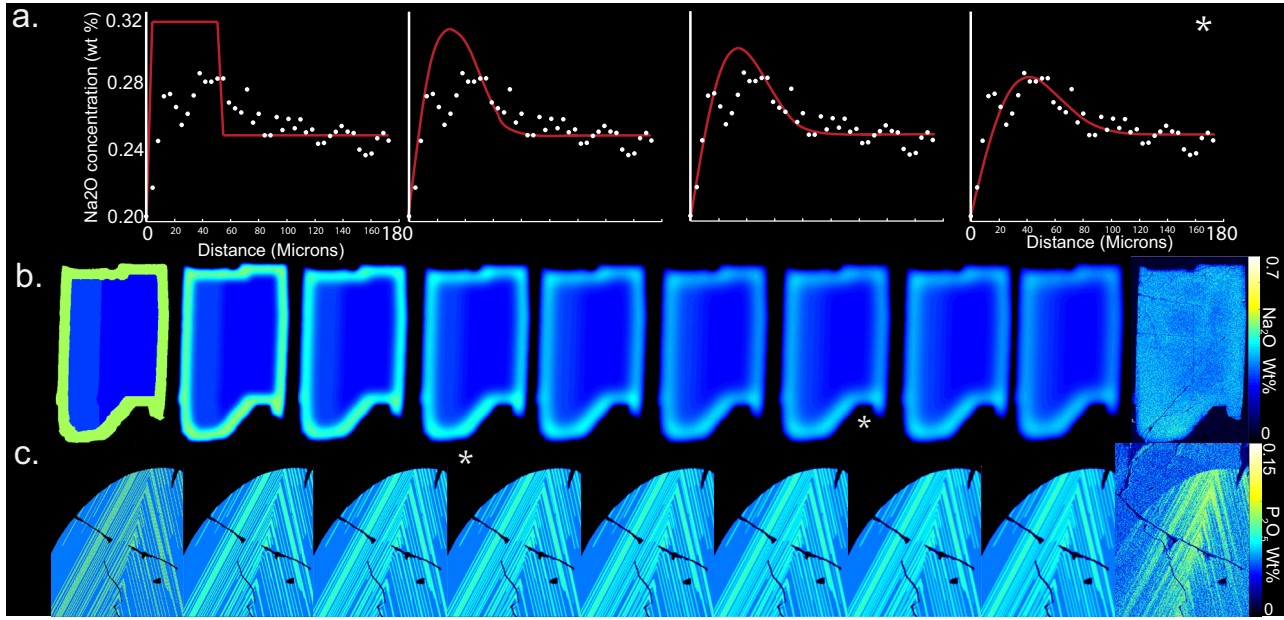

**Fig. 3 Graphical depiction of diffusion models. a** 1D on plagioclase in slide, 52 (Fig. 2b). **b** 2D on plagioclase in slide, 46. **c** 2D on olivine in slide, 46. Asterisk represents minimum RMSD.

**Table 1 Maximum calculated survival timescales for the observed compositional heterogeneities.**

|  | Plagioclase, 46 | Plagioclase, 52 | Olivine, 46 | | Olivine, 52 | |
| --- | --- | --- | --- | --- | --- | --- |
|  |  |  | $D_P$ | $D_{Si}$ | $D_P$ | $D_{Si}$ |
| Diffusion time (My) | 22.0 | 20.1 | $1.2 \times 10^{-5}$ | 27.1 | $9.5 \times 10^{-6}$ | 27.1 |

Slide numbers are given to represent the crystals described in Supplementary Fig. 5, with plagioclase, 52 referring to crystal Pl52_1

**Diffusion modeling**. Diffusion chronometry resolves timescales necessary to relax compositional heterogeneities diffusively by forward modeling of Fick's second law (Fig. 3), given appropriate laboratory-determined values for cation diffusivities ($D_P^{ol}$[32] and $D_{Ca-Na}^{plag}$[33]). The cooling path is the dominant variable in determining total diffusion timescales and need to be chosen carefully. To complete any diffusion modeling, we must assume an initial temperature and cooling path to obtain the total timescale over which diffusion occurred. A temperature of 1297 °C was used for olivine (liquidus temperature), and of 1229 °C for plagioclase (appropriate for the most sodic portion of the rim). Using these initial temperatures, a 3.9 °C My$^{-1}$ linear cooling path[22] results in complete homogenization of Na in plagioclase before the system cools by 0.2 °C. Phosphorus in olivine homogenizes by 1296 °C. Even if P diffusivity approaches that of slow diffusing, similarly coordinated elements[34] like Si[35], ~5 orders of magnitude lower than the published[32] $D_P^{ol}$, complete chemical homogenization of P is still attained by 1241 °C. Regardless of the diffusivity value chosen, the temperatures at which grains achieve chemical homogeneity are well above the diffusive closure temperature of Ca–Na in plagioclase and P in olivine. Therefore, it is necessary to consider faster, nonlinear cooling to explain the preservation of chemical zoning.

We sought the maximum survival time of these heterogeneities by iteratively testing various functions and fit coefficients over many millions of runs to impose nonlinear cooling paths. The boundary conditions are detailed in the "Methods" section; in summary, all cooling paths are set up to end between 800 and 900 °C where they connect with the linear cooling rate proposed by Borg et al.[22].

If phosphorus diffusion coefficients for olivine are calculated using the existing expressions[32], the timescales are extremely short (Table 1), six orders of magnitude shorter than indicated by Na–Ca interdiffusion in the neighboring plagioclase crystals. However, petrography indicates co-crystallization, such that each crystal in the rock provides an independent expression of the same cooling history. Seeking to remedy this inconsistency, we substituted $D_{Si}^{ol}$ for $D_P^{ol}$ in most olivine diffusion models. Like Si, P is a high-valence cation occupying tetrahedral sites in the olivine lattice[31,34,36]. The maximum cooling times allowable by the diffusion models are 20 and 27 My for plagioclase and olivine respectively. These cooling timescales are an order of magnitude shorter than previous estimates[22,24,37,38] (all above 100 My). Note that subsequent reheating events are included in this estimate. Thus, should reheating take place, initial magmatic cooling timescales would have to be even shorter than our estimates in order to preserve these heterogeneities.

Though these results contradict previous estimates for the early cooling history of troctolite 76535, they do not invalidate past measurements of the cooling rates in this sample. Magmatic cooling histories have previously been reconstructed from the lower-temperature portion of the overall cooling history[22,24,37]. Therefore, the cooling rates reported by each of these studies form a coherent but nonlinear single cooling path. Our results pertain to the highest temperature portion of this path (Fig. 4).

## Discussion

**Implications for the petrogenesis of troctolite 76535.** The discovery of sharp compositional gradients in the form of P-rich lamellae in olivine from 76535 (Fig. 1) imposes important new

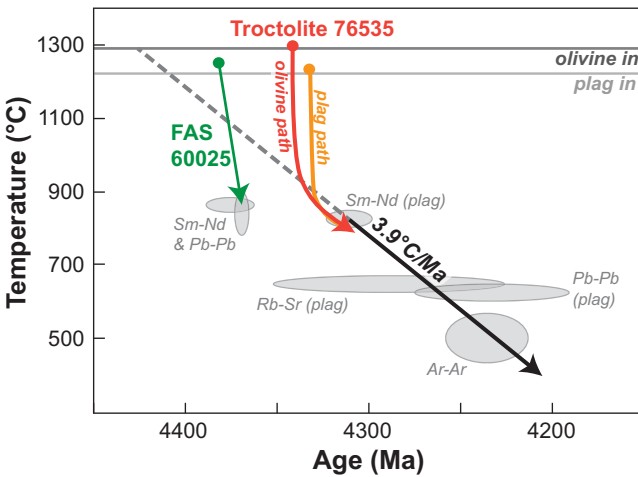

**Fig. 4 Colored cooling paths representing modeled maximum cooling time compared to linear cooling of 3.9 °C My$^{-1}$ (dotted line).** Neither solution conflicts with the formation of FAS 60025 (green line). FAS liquidus from Rapp and Draper[70] and cooling path from McCallum and O'Brien[65]. After Borg et al.[22].

qualitative constraints in addition to the quantitative maximum cooling timescales. A comparison with a terrestrial troctolite in slow-spreading oceanic crust at the Atlantis Massif[26,39] (previously mischaracterized as a gabbro[26]) illustrates these constraints. Like the olivine in 76535, high-Fo olivine grains in these terrestrial lower crustal rocks contain P enrichments[40,41]. The P-rich lamellae in lunar troctolite 76535 (Fig. 1) are much sharper than those preserved within the Atlantis olivine grains (Supplementary Fig. 4), suggesting more rapid overall cooling for 76535. The average magma cooling rate estimated using the massif assembly duration (0.2–2 My[40,42]) is 600–6000 °C My$^{-1}$, which in itself is rapid due to the tectonic exhumation of this lower crustal section along a long-lived normal fault. Thus, the petrographic comparison indicates that 76535 cooled through diffusion-active temperatures ≥2 orders of magnitude faster than previously suggested for 76535 (1–50 °C My$^{-1}$ (refs. [22,24,38])). Having established rapid initial cooling of the magma from which 76535 parent magma crystallized, we review prior observations and suggest a new conceptual model for the origin of the Mg-suite.

Mineral texture such as euhedral external morphology, concentric plagioclase compositional zoning, and poikilitic intergrowths require crystallization to have occurred initially from a liquid. The preservation of steep concentration gradients of phosphorus in olivine, and resolvable Ca–Na zoning in plagioclase, require relatively rapid cooling through magmatic temperatures. Relatively sharp, fine-scale lamellae of P in olivine further reveal that initial crystal growth was rapid[25,26,31]. Yet the presence of regions lacking P enrichments (Fig. 1) outside cuspate dissolution surfaces imply that olivine precipitation resumed at a relatively low growth rate. These new observations must be reconciled with indicators of a slow subsolidus (and thus below diffusion closure temperature) cooling rate: triple junctions at plagioclase grain boundaries and a three-point cooling rate line formed by radiometric ages[22].

Terrestrial troctolites are interpreted to have formed primarily by either accumulation of mineral grains within a magma body, or partial replacement of rock via reactive melt infiltration[41]. We consider these geneses in view of the thermal constraints mandated by 76535. The first scenario requires the existence of a magma body with a high volumetric proportion of liquid relative to solid phases. Recent experimental and numerical treatments have rekindled and refined two long-standing models for the formation of a

liquid-dominated body in the lunar interior. Melting may have occurred at the boundary between overturned mantle cumulates and FAS crust units in the vicinity of (and possibly augmented by) a liquidus-lowering ur-KREEP component[12,13]. Alternatively, liquid may have accumulated as the result of decompression-driven partial melting associated with the overturn itself, and not require a hybrid source[6]. A third possibility is that the liquid-rich magma body formed via impact[14,15]. In all of these cases, formation of troctolite occurs as the liquid lens progressively crystallizes to form a layered sequence of lithologies (c.f. the lower units of the Kiglapait or Skaergaard intrusions[43,44]). Our results inform the discussion of Mg-suite petrogenesis chiefly through considering the implications for the thermal history. Importantly, a liquid-rich magma body capable of generating layered cumulate lithologies is likely to be spatially compact and coherent over length scales of 100's of m to 10's of km. The maximum size of a spherical magma body consistent with the modeled magmatic thermal histories has a radius of 11 km (Methods, Supplementary Fig. 8). A more realistic lenticular magma body could be of higher volume and still satisfy the initial rapid cooling criterion but would then cool too quickly (internally) to satisfy the subsolidus slow-cooling criterion.

A second scenario is that the troctolites form via reactive melt infiltration, (Fig. 5). This model differs from those above in that the system is solid-dominated rather than liquid-dominated. Within the context of this solid-dominated system, our results make it possible to consider, in greater detail, the physical picture of troctolite generation from the hybrid source[12,13]. Hybridization requires that the crust behaves as a plastic solid upon interacting with rising lower mantle material. Mafic cumulates could mechanically mix with the lower crust, permitting the two lithologies to mingle chaotically and creating intimate mixtures characterized by high inter-lithologic contact area over small (cm to m) length scales. Cotectic melting would thus occur over a widely distributed volume comprising the former lower crust and upper mantle, but melt would accumulate locally to form plexuses of melt-rich lenses that promote reactive melt infiltration[41,45–47]. The hybrid (LMO olivine and FAS anorthite) would then partially melt and interact with ur-KREEP, forming a reactive liquid saturated in forsteritic olivine[48] as well as anorthite. Unsteady melt production and pulsatory porous flow during the interaction of the mingled parent lithologies could produce the cycles of local olivine and plagioclase dissolution-reprecipitation that are required by the new observations we present.

As with Atlantis Massif units, this sequence of events could be expected to produce all lithologies found in the Mg-suite[49]. In the chaotic mixing of cumulates, mixtures ranging between pure mafic cumulates and pure crustal material would be created. Melts in these different zones would have a range of compositions, and crystallize different products, potentially producing several Mg-suite lithologies. In addition, reactive melt that interacts with the surrounding crust could create the pink spinel anorthosites[50]. The KREEP-poor members of the Mg-suite like meteorites[51] could also be explained in this scenario; should ur-KREEP have not been evenly distributed beneath the lunar crust[1,13], the reactive infiltration process would create KREEP-poor Mg-suite samples as posited for regions outside of the PKT.

The conceptual description above presumes the Mg-suite parent is a hybridized source. Formation of the Mg-suite parent liquids by decompression melting of mafic cumulates[6] is also compatible with the reactive melt infiltration process, as demonstrated by the existence of olivine-rich troctolites of the Atlantic Massif that owe their existence to iterative reaction between partial melts of Earth's mantle and lithospheric peridotite. In fact, formation of the suite of the lunar Mg-suite lithologies by progressive reaction of mantle-derived liquids with just-formed lithologies may pose fewer challenges than the

**Fig. 5 Petrogenetic model for troctolite 76535 involving reactive melt transport. a** Mg-rich lower mantle interacts with a warm anorthitic crust, and ur-KREEP, mechanically mixing the three components. **b** Partial melting occurs; the liquid infiltrates the surrounding rock and accumulates in lenses (cm–m scale). **c** Microscopic views of rapid initial crystallization followed by dissolution and crystallization of plagioclase (gray), olivine (green), and orthopyroxene (dark green) during rapid cooling. As the magma temperature approaches the temperature of the surroundings, the cooling rate slows. Grains anneal during protracted subsolidus cooling.

hybrid-source model, as it requires involvement (physical juxtaposition) of fewer physically distinct sources.

We suggest that the preferred scenario involving reactive melt infiltration presents two advantages over cumulate layering. First, in the context of the hybrid-source model, the fine (cm) length-scale interaction among FAS and lower mantle crystals would promote cotectic melting, which is energetically favorable because interphase grain boundaries melt at much lower temperatures than pure end members[52]; the exorbitant heat budget required to melt FAS is a long-standing problem in Mg-suite genesis[53]. Second, and relevant to both source-melt models, reactive infiltration provides a compelling explanation for the observed textures. The olivine P-enrichment patterns in 76535 require rapid crystal growth, dissolution, and lastly slow re-growth, a cycle that is more compatible with solid-dominated than liquid-dominated systematics.

Powerful constraints of Mg-suite petrogenesis have been provided by a detailed analysis of troctolite 76535. However, to truly test these findings similar analysis must be completed on more samples from the Mg-suite in the Apollo collection, and magnesian suite like meteorite clasts (especially a comparison of KREEP rich and KREEP-poor clasts, exploring the thermal histories of these two suites, which would give insight into the role of KREEP signatures in the petrogenesis of the Mg-suite). Of special interest to the question of petrogenesis is an analysis of the different major olivine and plagioclase bearing rock types of the Mg-suite, such as dunites (prime canidates: 72415–72418), spinel troctolites (prime canidate: 67435), norites (15360), and perhaps gabbronorites (should any suitable grains be found). The data from such studies could provide significant new constraints on Mg-suite petrogenesis and the evolution of the lunar crust. Given the recent discovery that many samples of the Mg-suite formed nearly contemporaneously[54], it may be possible to obtain a cooling history for the rest of the suite as well.

Resolving the timing of important events in the Moon's history is at the frontier of lunar science. The Moon-forming giant impact (GI) is suggested to have occurred "early and old" relative to the formation of the solar system[55], or "late and young"[23]. Combined U–Pb and Lu–Hf isotopic systematics of lunar zircon crystals are only consistent with the older GI ages and they suggest that the LMO solidifed rapidly[56]. These old zircon ages contrast with the slower time frame of LMO solidification[57] proposed to resolve the paradox posed by coincident model ages for the FAN and Mg-suite[23], as well as the considerably younger Sm–Nd model ages obtained for the Mg-suite samples[22].

Our data set is not capable of critically evaluating the GI ages or the duration of LMO overturn. However, our observations do refute the suggestion that the young Mg-suite Sm–Nd ages are an artifact of very slow cooling from the magmatic liquidus through the low closure temperature of the Sm–Nd system[56]. Our results support rapid cooling through the Sm–Nd closure temperature in 76535 and by extension, contiguity of the model Sm–Nd ages with magmatic ages for the Mg-suite.

Preservation of sharp P-rich lamellae impose thermal constraints for future dynamic modeling. If the troctolite parent liquid crystallizes in an area dominated by hot lower mantle, then it would cool too slowly at subsolidus temperatures for relatively sharp P-lamellae retention. Hence, 76535 poses a goldilocks scenario. The proportions of the dominant thermal masses (hot lower mantle cumulates and warm anorthite) and the intimacy of their mixing in the melt source region, must have been just right.

## Methods

### Grain selection

*Plagioclase*. Na–Ca interdiffusion in plagioclase may be modeled in grains that: (a) preserved igneous growth facets, (b) were isolated from phases that possess or accept Na, (c) possessed minimal cataclastic features such as cracks and brecciation. One grain lacking requirements (b) and (c) was mapped and found to be homogeneous with respect to sodium, suggesting grain contact limited preservation of heterogeneities (Supplementary Fig. 2). Thus, we regard all conditions (a–c) as requirements for application of the numerical diffusion chronometry technique.

Despite the large proportion of plagioclase in this sample, very few grains match all three criteria. We inspected six slides of troctolite 76535 that contained hundreds of plagioclase grains. Of these grains, four were suitable for diffusion chronometry. All four grains are surrounded by either olivine or orthopyroxene, preserving igneous growth facets.

*Olivine*. The skeletal structure of phosphorus enrichment patterns in olivine removes the growth facets condition as well as the isolation condition. The high pristinity of this sample ensures that virtually any grain containing P heterogeneities is suitable for modeling. From the crystals we mapped at low magnification, we selected two nested areas for high-resolution mapping to be used for diffusion modeling.

**Analytical methods**. Quantitative profiles and qualitative X-ray intensity maps were acquired using the University of Hawai'i at Manoa's JEOL JXA-8500F field emission electron microprobe.

*Olivine*. Olivine profiles in Mg, Fe, Si, and P were collected using a 15 kV accelerating voltage, a 200 nA beam current, 1-µm beam size, and 90 s on peak time. The background was modeled using an exponential fit using Probe for EPMA v.12.7.2 (PfE). Phosphorus was measured on two spectrometers using the "aggregate intensities" function on PfE to increase analytical precision. Following precedent for the study of phosphorus in olivine[25,58], an apatite standard (USNM 104021) was used for phosphorus, San Carlos USNM 111312/444 and Springwater

USNM 2566) with concentrations of phosphorus known to be below the detection limit of our instrument were used as blanks, corrected by PfE. These two olivine standards are also used for Fe, Mg, and Si calibration. An A-99 Basaltic glass standard and Springwater USNM 2566 were analyzed as unknowns to track drift and accuracy (Supplementary Table 2). No drift correction was necessary, and standard analysis remained accurate throughout measurement.

Olivine maps were collected with a 15 kV accelerating voltage, a 200 nA beam current, a 100 ms dwell time, a step size between points of half a μm for high-resolution maps used for modeling, up to 3.5 μm for overview maps, a focused beam was used. All five spectrometers were set to measure phosphorus.

*Plagioclase.* Analytical profiles of Na, Si, Ca, Fe, Mg, and K in plagioclase were collected with a 15 kV accelerating voltage, beam currents of 40–100 nA, and a beam diameter of 4 μm. On peak times of 60 s were used for all modeled profiles.

Compositional maps were produced under similar conditions, though conditions varied by grain. The plagioclase grain in slide 46 is greater in size, permitting a larger beam diameter. For this grain, the following conditions were used for mapping: 15 kV accelerating voltage, 200 nA probe current, a 3-μm beam diameter, a 90 ms dwell time, and a 1.40-μm step size. The rest of the maps were collected over a smaller region, necessitating different conditions: 15 kV accelerating voltage, 125 nA probe current, focused probe diameter, a 90 ms dwell time, and a 0.60-μm step size. An anorthite standard (NMNH 137041) was analyzed as an unknown to track drift and accuracy (Supplementary Table 3). No drift correction was necessary for Na in either slide used for modeling.

**Diffusion modeling.** The presence of compositional heterogeneities in these samples can be leveraged to quantify maximum allowable timescales of diffusion during cooling. In order to model diffusive relaxation, boundary conditions appropriate for the two different diffusive systems are required.

*Olivine set up.* The initial phosphorus distribution in the 2D olivine maps were assumed to be sharp and drawn as smaller than or equal to two pixels wide (Figs. 1 and 3). Choosing thin phosphorus zones results in longer diffusion timescales than thick zones. We assumed very thin initial P enrichments in order to recover the maximum duration of cooling. Moreover, high-resolution phosphorus maps reveal phosphorus lamellae are on the order of a μm thick (Fig. 1).

Lacking independent constraint on the initial concentrations of phosphorus (in addition to the original lamellar thickness, as explained above) we again choose based on the goal of maximizing the retrieved diffusion time. In this case, we take the maximum observed concentration from the high-precision profiles to be the initial P concentration in lamellae. The maximum observed value must be lower than the initial value, if any diffusive homogenization has occurred. This underestimation in the initial concentration serves to maximize the calculated diffusion times, as, if all other variables are held constant, a lower concentration gradient leads to lower diffusive flux (i.e. Fick's first law).

*Plagioclase set up.* The initial conditions in plagioclase were set up to simulate the complex "shoulder-shaped" Na zoning after diffusion starting with a normally zoned grain. This normal zoning was, for simplicity, modeled as a step function in Na content (Figs. 2 and 3). This maximizes the diffusion timescale: as explained above. The thickness of the rim was determined by trial and error to be the minimum necessary to match the observed shape of the compositional gradient. This choice also maximizes the calculated diffusion times, as thicker bounds require less time for diffusive homogenization to occur. A semi-open boundary on the outside of the grain that permitted a degree of diffusive escape, creating the "shoulder" pattern.

One grain on slide 46 contains a region of exsolved pyroxene on the left side of the grain. Sodium content in this area is generally higher than the surrounding portions of the grain. Therefore, the right side of the grain more accurately preserves the pre-exsolution concentration gradient. The model was therefore optimized to favor the right side of the crystal. This was done by adding a zone of higher sodium content in the interior of the crystal on the left side (Figs. 2 and 3). In effect, this reduces the deviation on the left side of the grain, weighing diffusive timescales to be affected more by the region outside of the exsolution zone.

The X-ray map was calibrated using Na concentrations measured along an analytical profile through the width of the crystal. The average Na concentration at the core of the grain was chosen as the concentration for the interior of the grain in modeling. The value of the rim was then picked by finding the total sodium content above the background concentration. This was done using Simpson's rule for Riemann Sums, which is a method of area approximation using a sum of boxes with caps approximating the curve using simple polynomials. Once the total sodium content above background is known, the sodium density of the rim is fixed so the sodium content is invariant between the model and the data.

*Initial temperature.* To define initial temperatures, it is necessary to characterize the composition of the parent melt for this system. We attempt to recreate the parent melt using the trapped melt composition of Mg-norite 78235,47 (ref. [59]), combined with O'Sullivan and Neal's[53] constraints on 76535 parent melts. Finally, the composition was altered to produce the magnesian olivine and calcic plagioclase

compositions reported for this sample[60] using rhyolite-MELTS[61,62]. The resultant liquid composition is reported in Supplementary Table 1.

The mineral-in temperatures for both olivine and plagioclase are calculated using rhyolite-MELTS[61,62]. All calculations utilized an oxygen fugacity corresponding to one log unit below the iron–wüstite buffer (IW-1), in order to match the $fO_2$ inferred from lunar basalts[63]. Magmas are nominally anhydrous on the Moon[64]; hence, phase appearance temperatures increase with pressure (Supplementary Fig. 1). Diffusion rates increase with temperature, so a pressure of 1 bar was used to maximize diffusion times. If this sample formed deep in the crust, it would have experienced more rapid magmatic cooling than determined here to accommodate the faster diffusion associated with higher crystallization temperatures. The MELTS models predict mineral-in temperatures of 1297 °C for olivine, and 1244 °C for plagioclase. These values are used as initial temperatures in the diffusion models.

Diffusive relaxation can only begin after the sodic portion of the grain forms. Therefore, to ensure conservative timescales, we chose a starting temperature associated with the most sodic portion of the rim as calculated by MELTS. A concentration gradient must have existed before this temperature, and our initial bounds will start at this lower temperature, essentially both delaying the start of diffusion, and slowing the diffusion that does occur.

As no systematic variation in Mg is detected in the olivine grains, this procedure to lower the starting temperature from the liquidus could not be completed for those crystals, and the liquidus temperature was used as the initial temperature. Skeletal olivine morphology is a hallmark of rapid crystal growth, supporting the validity of this approximation.

*Diffusion.* We model diffusion using both profiles and X-ray intensity maps collected on all samples as output constraints. Fick's Second Law for Diffusion (Eqs. 1 and 2) was solved using finite differences in the forward direction.

$$\frac{\partial C}{\partial t} = D * \frac{\partial^2 C}{\partial x^2}$$

Equation 1: Fick's second law for diffusion in the one-dimensional case

$$\frac{\partial C}{\partial t} = D \Delta C$$

Equation 2: Fick's second law for diffusion in the *n*-dimensional case, where $C$ = the concentration at any given point in wt%, $t$ = time [s], $x$ = position [μm], and $D$ is the diffusivity [μm$^2$ s$^{-1}$] defined in Eq. (3).

$$D = D_0 e^{-\frac{Q}{RT}}$$

Equation 3: Diffusivity calculation for the systems utilized here

Here $T$ = temperature [K], $R$ = the gas constant [m$^3$ Pa K$^{-1}$ mol$^{-1}$], $D_0$ is the pre-exponential factor [m$^2$ s$^{-1}$], and $Q$ is the activation energy [m$^3$ Pa mol$^{-1}$]. Values for $D_0$ and $Q$ are from Dohmen[35] and Watson et al.[32] for olivine and Grove[33] for plagioclase. Temperature is recalculated at each time step to follow the cooling path being considered decreasing from the initial temperature for the crystal.

*Fit coefficients for cooling paths.* Fit coefficients were found in the form $T = T_0 - a*f(b*t)$, where $T$ = temperature, $T_0$ = initial temperature, $t$ = time, and constants $a$, $b$ that we wish to find. Only a certain range of $a$ and $b$ coefficients will produce a cooling function that is continuous and differentiable with respect to the smooth-transition condition (3.9 °C My$^{-1}$ cooling at 850 °C after Borg et al.[22]). The range of acceptable $a$, $b$ coefficient pairs were found by repeated diffusion modeling on each crystal. The temperature and temperature rate of change were then plotted as functions of $a$ and $b$ (Supplementary Video 1). These plots were then used to guide the choice of the next pair of coefficients, with the aim of interpreting the shape of the two surfaces and finding where they constitute solutions to the smooth-transition condition. Of this range of acceptable values, the solution that provides the maximum diffusive timescale is chosen as the representative function. This is very computationally intensive, and hundreds of thousands to millions of calculations were completed for each crystal, requiring the scripts to be parallelized, and run on the University of Hawaii's High-Performance Compute Cluster, as well as multiple personal computers.

**Thermal modeling.** We construct a simple thermal model to determine the maximum size of a magma chamber that could cool within 20 My, the maximum duration permitted by our thermal modeling of the compositional heterogeneities. The thermal model assumes a spherical, radially symmetrical chamber emplaced into a cool anorthosite crust. This is solved, as for the case of chemical diffusion, by application of finite differences to Fick's second law. In this case:

$$\frac{dT}{dt} = \frac{k}{\rho * c_p} * \frac{dT^2}{d^2 r}$$

where $T$ = temperature [K], $t$ = time [s], $k$ = the thermal conductivity [W (m K)$^{-1}$], $c_p$ is the specific heat (J (kg K)$^{-1}$), and $r$ = radius [m]. A cooler crust would permit a larger magma chamber to cool within a given time frame. As crustal temperature could be highly variable and we have no independent constraints, we choose a temperature of 650 °C as an extrapolation of the 18 °C My$^{-1}$ cooling path[65]. Should the crust have been cooler, a larger magma body would be possible. The

initial melt temperature of 1297 °C was the same as used in the olivine diffusion models. For the crust, a $k = 1.76$ W m$^{-1}$ K$^{-1}$, $\rho = 2730$ kg m$^{-3}$ for anorthosite[66], and a $c_p = 716$ K kg$^{-1}$ K$^{-1}$ of anorthite[67]. Parameters for the initial melt were $k = 1.09$ W m$^{-1}$ K$^{-1}$, $\rho = 2993$ kg m$^{-3}$, and $c_p = 1700$ kg$^{-1}$ K$^{-}$ for an olivine-melilitite melt[68]. Once the temperature dropped below 1229 °C, the thermal parameters were switched to troctolitic values[69]: $k = 3.06$ W m$^{-1}$ K$^{-1}$, $\rho = 2850$ kg m$^{-3}$.

**Electron backscatter diffraction**. Crystal orientation for five grains on slide 76535, 46 and 76535, 52 was collected using electron backscatter diffraction (EBSD). This technique was completed on a JEOL 5900LV scanning electron microscope operated at low vacuum (11 Pa), using an Oxford Instruments HKL Channel 5 EBSD detector. The slides were prepared by light polishing to remove the existing carbon coat. The prepared surface was sufficiently smooth that beam interaction with the sample surface produced indexible electron backscatter diffraction patterns (EBSPs) without colloidal silica polishing. The samples were tilted at a 70° angle relative to the 15 kV electron beam.

Working distances (and thus beam currents) were minimized given the constraints of our instrument's detector geometry. For the olivine in slide, 46, a 20 mm working distance was used for the grain in the bottom right of Supplementary Fig. 7, and an 18 mm working distance was used for the two points on either side of the truncation. For the plagioclase in slide, 56, a 15 mm working distance was used. Finally, for the two points in the olivine an plagioclase in slide, 52, a 17 mm working distance was used. Background was subtracted using the in-software image enhancement tools in Flamenco acquisition module of Channel 5 (Oxford Instruments) software; the binning, noise reduction, and background frame integration and timing-per-frame values were tuned manually to minimize the acquisition time required to obtain indexible patterns, i.e., EBSPs matching the solutions generated by phases in the American Mineralogist library. As 76535 is a very coarse-grained sample, background patterns were usually collected either using a symplectite, or at junctions with more than five grains in view. Band detection and indexing conditions (Hough resolution, number of reflectors) were tested iteratively on live EBSPs to ensure proper indexing in automated mapping jobs.

To collect the data, a 12 × 12 map was made on each grain. Step size was varied between maps to avoid cracks and any surface debris. The data were post-processed and displayed using the Channel 5 modules Tango (misorientation maps) and Mambo (pole figures).

## Data availability
The EPMA data generated in this study have been deposited at figshare.com at https://figshare.com/projects/Chemical_Heterogeneities_Reveal_Early_Rapid_Cooling_of_Apollo_Troctolite_76535/120165. For other data formats or any questions, please contact William Nelson at wnelson@hawaii.edu.

## Code availability
Diffusion modeling code is available upon request, please email wnelson@hawaii.edu.

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

## Acknowledgements

The authors thank G. Huss and A. Kirsch for thoughtful discussion on an early version of a manuscript, and Robert Nelson for assistance with stitching of XPL context photographs. The technical support and advanced computing resources from the University of Hawaii Information Technology Services—Cyberinfrastructure are gratefully acknowledged. Funding for this research was provided by NASA award NNX16AO77G to J.H. This is SOEST contribution 11407.

## Author contributions

J.E.H., T.S., E.H., and G.J.T. conceptualized the project. W.S.N. and T.S. obtained data. W.S.N. modeled heterogeneities. All authors interpreted results. W.S.N. and J.E.H. wrote the paper.

## Competing interests

The authors declare no competing interests.
