## [Peer Review File · Nature Communications]

REVIEWERS' COMMENTS

Reviewer #1 (Remarks to the Author):

Nelson et al., present chemical analysis of and diffusion modeling for Apollo troctolite 76535 in order to further constrain petrogenesis of the enigmatic highlands Mg-suite. The authors find Phosphorus (P) heterogeneities in olivine, which are consistent with terrestrial analogs and indicative of rapid cooling from magmatic temperatures. Modeling results and analog comparisons suggest that sample 76535 cooled at a rate >2 orders of magnitude faster than previous estimates. The authors then synthesize their findings with some previous models of origin and discuss qualitative implications concerning the existing package of petrogenetic models.

In general, I find that the authors have conducted a detailed chemical analysis of 76535 and extract new data on one of the most well-studied samples in the Apollo collection. The analysis and diffusion modeling presents significant compositional, temporal, and thermal constraints for the origin of the Mg-suite. Because the formation of Mg-suite is intimately tied to constraining the guiding paradigm of lunar differentiation (lunar magma ocean), the topic is well-suited for the readership of Nature Communications.

However, I find that the manuscript undersells its important compositional, temporal, and thermal constraints presented and as a result, overreaches with respect to some of its qualitative comparisons. Unfortunately, a rapid shift to the qualitative comparisons raised a number of quantitative questions regarding the proposed model, and also how the new results factor into previous conclusions. In providing more detailed analysis of their findings, the discussion should follow a systematic evaluation of the current models. Here, each model of origin should be filtered through the same set of chemical, temporal, and thermal constraints defined by their measurements and modeling in determining which hypotheses require further attention or which hypotheses can potentially be ruled out on this basis.

Below, I've outlined five considerable considerations related to the compositional, temporal, and thermal constraints presented. These are followed by some relatively minor comments related to section/line aspects in the manuscript. After careful consideration, I recommend that the paper be returned to the authors in order to address these comments for revision. I have selected topics of discussion that I believe will strengthen the results by framing into a more quantitative and comparative context, help broaden the current scope, and present the reader with important and relevant findings of fact for future considerations and further scrutiny of the present data set.

- 
1. The results motivate reexamination of other olivine-bearing Mg-suite samples in order to identify their cooling rates based on the methods and P-analyses presented. Thus, the results should be incorporated into a more comprehensive cooling history for Mg-suite rocks in general (or to understand any variances among location and landing sites). This motivation is not directly/explicitly expressed. I find this to be a primary topic and implication of the study in and of itself, and the manuscript should not shy away. I find that it is definitely worth reviewing/noting up front since it **a)** forwards a method for adding value to the existing Apollo collection in addition to **b)** providing a means to further scrutinize the conclusions of the manuscript.
 - a. Importantly, the manuscript does not revisit the previous models that proposed much slower cooling histories. How do the new and faster cooling rates support/contradict the previous findings of (e.g.,) exsolution and ordering in pyroxenes? Can the exsolution lamellae and ordering be explained with the faster cooling rates, or do they still require the slower cooling rates? If the latter, how are these two issues reconciled?
2. All throughout, the manuscript appears to favor a hybridized source model which requires that primary mantle cumulates mixed with crust + KREEP prior to then partially melting to form the KREEP-bearing cotectic Mg-suite parental melt. After careful review, some of the data presented in the current manuscript appears to contradict the present conclusions. Further, the manuscript should provide more context to the issue of petrogenesis such as KREEP-poor meteorites, previous experimental work testing the hybridized source model, and orbital remote sensing. It is for these reasons a more critical analysis is necessary in order to justify the present conclusions or incorporate important amendments during revisions.
 - a. The maximum reported abundance of P₂O₅ as measured by the author's 5-spectrometer method is ~ 0.1 wt.%, and with a minimum of 0.006 wt.% P₂O₅ (noting that this range is defined by a single profile measured on a single crystal of olivine). Building upon your terrestrial analog (and to my knowledge), the present 76535 P-values are consistent with P-measurements in olivine for many mantle-derived terrestrial samples (0.006 – 0.2 wt.% P₂O₅; Milman-Barris et al., 2008). The P-in-olivine correlations between 76535 and mantle-derived terrestrial samples seems to simply suggest that 76535 was also derived from mantle-melting (e.g., Longhi, 1981; Prissel and Gross, 2020), and that there is no need for a complex hybridized source. Can you demonstrate that similar P-abundances in olivine are expected in both the terrestrial mantle-derived samples and Mg-suite troctolites derived from a hybridized KREEP-rich source?
 - b. How does the proposed hybridized source model satisfy KREEP-poor and KREEP-rich magnesian-suite-like meteorite clasts (e.g., Gross et al., 2020) originating from source localities outside of the PKT?
 - c. Another important constraint from orbital remote sensing: the hybridized source model suggests olivine cumulates are transported to the base of the lunar crust. But orbital remote sensing indicates that olivine mantle exposures in large impact basins are rare (e.g., Yamamoto et al., 2010; Melosh et al., 2017; Moriarty et al., 2020).

Prissel and Gross (2020) circumvented this issue by suggesting that Mg-suite mantle-melts may have been extracted in a similar manner to picritic glasses. How does the proposed hybridized source model reconcile the paucity of olivine-mantle exposures in the remote sensing database?

3. The manuscript presents diffusion modeling which constrains the cooling timescales of Mg-suite to $< 27\text{My}$ (or $< 20\text{My}$ based on olivine alone). If so, this severely limits the expected or anticipated range of crystallization ages of Mg-suite samples from the isotopic system of interest.
 - a. Borg et al., (2020) report the Mg-suite on a whole crystallized $\sim 4340 \pm 9\text{Ma}$. This tightly constrained formation age of Mg-suite is not necessarily expected if the cooling timescales for some Mg-suite samples in the crust are capable of exceeding 100My time prior to isotopic closure. I think this is an important result that adds relevant discussion to temporal issues of Mg-suite in addition to providing further context to the more confounding issue in lunar science – the near-contemporaneous relationship between LMO-products and Mg-suite.
 - b. Geodynamic models suggest the onset of mantle overturn can occur near-contemporaneously with LMO solidification (e.g., Boukaré et al., 2018; Li et al., 2019), and so Mg-suite models directly tied to mantle overturn appear to satisfy this issue. The decompression-melting model proposed by Prissel and Gross (2020) demonstrated that partial melting of primary LMO cumulates as a direct result of mantle overturn can produce successful KREEP-free Mg-suite primary melts. The hybridized source hypothesis is not as direct or straight forward, and it is unclear how quickly mantle cumulates can mix in the precise proportions to produce successful Mg-suite melts as presently defined (it is only assumed to be instantaneous relative to the onset of mantle overturn, but this is not quantified – Shearer et al., 2015). The present manuscript is not necessarily equipped to address the unquantified time constraints associated with the hybridized source hypothesis, but these are important aspects that deserve attention in the discussion.
 - c. Coupled U-Pb and Lu-Hf studies suggest rapid LMO solidification and older LMO model ages (e.g., Taylor et al., 2009; Barboni et al., 2017; Crow et al., 2017). In particular, can your interpretations of rapid Mg-suite crystallization be reconciled with $^{207}\text{Pb}/^{206}\text{Pb}$ crystallization ages of lunar zircon and Lu-Hf model ages of LMO solidification at $\sim 4.51\text{Ga}$ (Barboni et al., 2017), followed by mantle-overturn-induced Mg-suite $\sim 200\text{My}$ later at 4.34Ga ?
 - d. Some further considerations: see also the slow-cooling magma ocean model recently proposed by Maurice et al. (2020) that is consistent with an initial LMO solidification age of $4425 \pm 25\text{Ma}$, which predates the formation of Mg-suite and suggests that the younger formation age of 60025 ($4359 \pm 9\text{Ma}$; Borg et al., 2020) is unrepresentative of the earliest formed crust and instead may represent lunar crust forming at the final stages of a slow-cooling magma ocean. In this way, Mg-suite crystallization ages do not necessarily pre-date the formation of all lunar crust.
 - e. The authors may find it also relevant to add discussion on the complications associated with interpreting the age data itself including low-closure temperatures for the Sm-Nd isotopic system and thus, potential isotopic resetting events such as mantle overturn (or Mg-suite magmatism itself), KREEP metasomatism (if operative), giant basin forming impacts, radioactive heating within the Procellarum

- KREEP Terrane, or a possible combination of these (e.g., Borg et al., 2011, 2017; Barboni et al., 2017; Pernet-Fisher et al., 2019; Xu et al., 2020).
4. The paper presents a nice terrestrial analog for discussion purposes as it relates to a cumulate origin. However, the terrestrial analog contains Cr-spinel and this is an important petrogenetic indicator (e.g., Irvine 1965, 1967; Dick and Bullen 1984; Allan et al. 1988; Kamenetsky et al. 2001), particularly as it pertains to the Mg-suite (Prissel et al., 2016).
 - a. Prissel et al. (2016) experimentally tested the hybridized source model – summarizing: partial melting of the hybridized source yields troctolitic parent melts (olivine+plagioclase saturated; Longhi et al., 2010) as modeled here (Supplementary Table 1) with > 15 wt.% Al₂O₃. However, Prissel et al., (2016) experimentally demonstrate why troctolitic/cotectic magmas are potentially problematic as Mg-suite parental melts. The reason being is that troctolitic parent melts stabilize pink spinel owing to the high MgO and Al₂O₃ starting concentrations. Prissel and Gross (2020) point out that pink spinel troctolites represent < 2 wt.% of all lunar troctolites, whereas common lunar troctolites (> 98 wt.% of all lunar troctolites) contain magmatic Cr-spinel or evidence of magmatic chromite (or no pink spinel; e.g., Albee et al. 1974; Dymek et al. 1975; Lindstrom et al. 1984; Shearer et al. 2015).
 - b. Cr-spinel is of course ubiquitous among mantle-derived olivine-rich samples (see also the terrestrial troctolite discussed from the Atlantis Massif). Drawing from earlier suggestions (Warren, 1986; Ryder, 1991), and results from melt-anorthite reaction experiments (Prissel et al., 2014), Prissel and Gross (2020) concluded that minor amounts of pink spinel can be explained from minor amounts of assimilation of the lunar crust. The minor amount of assimilation required to produce minor amounts of pink spinel is consistent with anorthosite assimilation being a minor aspect (and likely not always operative) of petrogenesis as defined by experimental petrology and thermodynamic modeling (e.g., Morgan et al., 2006; Treiman et al., 2019). If the hybridized source model is the predominant mode of origin and yields troctolitic (or cotectic) melts, why then are pink spinel troctolites so extremely rare (Prissel et al., 2016)? Or put another way, why is the occurrence of magmatic Cr-spinel (or evidence thereof and lack of pink spinel) in > 98 wt.% of all lunar troctolite samples *not* indicative of a mantle origin (Borg et al., 2020; Prissel and Gross, 2020)?
 5. The manuscript offers an example result if the crust were cooler than their assumed 650°C input for modeling, and suggests no independent constraints are available. However, independent modeling of the thermal evolution of the lunar crust suggests at the time of Mg-suite magmatic emplacement the temperatures of the lunar crust could be hotter, near 800°C (e.g., Solomon and Longhi 1977; Toksoz et al., 1978; Warren, 1986). Moreover, Warren (1986) notes that models of early lunar thermal evolution did not consider the insulating effects of a thick megaregolith, which could've further increased local temperatures by ~100°C. In a figure of choice, the manuscript would be strengthened by quantitatively evaluating the effects of a cooler and hotter crust.
-

6. **Title:** The title is nondescript relative to its content. The authors should reconsider working in some common and descriptive search terms of their liking such as ‘lunar troctolite,’ ‘phosphorus,’ ‘diffusion,’ and/or the sample number 76535.
 7. **Abstract:** I recommend adding some quantification/results to the abstract. For instance, the element phosphorus is not even mentioned, but it is the cornerstone of this work. What are the cooling rates, and how different are they than previously estimated? Adding this information from the present manuscript will provide the glancing reader valuable and enticing information to dig deeper.
 - a. The authors may want to also reconsider the implications stated in the abstract after addressing the comments below.
 8. **73:** In general, try to avoid using the qualifier “all” when citing only a fraction of all studies.
 9. **79-80, 118-119:** The results would be strengthened by demonstrating in a figure that the oscillatory P-zoning you measured is independent of normal major element zoning (Mg/Fe) over the same region. And then describe in some more detail why previous work/experiments suggest rapid crystallization vs. other hypotheses such as compositional-fluxes during magma mingling.
 10. **85-100:** Previous studies have identified sub-solidus re-equilibration between olivine-spinel as an operative mechanism for Mg-suite samples (e.g., McCallum and Schwartz, 2001; Prissel et al., 2016). In particular, Prissel et al. (2016) demonstrate that pink spinel troctolites appear in olivine-spinel magmatic equilibrium, whereas Cr-spinel bearing Mg-suite samples appear to have experienced some Fe/Mg sub-solidus re-equilibration. These results were used to suggest pink spinel troctolites (cooled rapidly at the chill margins of a magmatic intrusion) whereas common lunar troctolites cooled more slowly at sub-solidus temperatures. Although the current study focuses on magmatic cooling, it might be worth mentioning to what extent (if any) we should expect P-heterogeneities that were locked in during rapid magmatic cooling to be later effected during sub-solidus heating events.
 11. **303-309:** It is stated that the parent composition in Table 1 was, in part, calculated based on the melt inclusion found in a lunar norite. Can you justify why using an evolved melt inclusion in equilibrium with orthopyroxene and plagioclase is appropriate for estimating a parental melt that first produced dunite → troctolite prior to norite assemblages? Why is this composition more appropriate than testing previous estimates of the Mg-suite parental melt (e.g., Longhi, et al., 2010; Prissel et al., 2016; Prissel and Gross, 2020)? Please provide more clarification as to how you determined the starting P and K abundances, or why they are assumed.
 12. **Figure 4:** Lin et al⁴⁷ is referenced but not listed in the bibliography. If the authors are referring to any one of the LMO experimental studies from Yanhao Lin, please be advised that this series of papers did not use Na or K (or P) in the starting compositions (i.e., as it pertains to FAS compositions and liquidus temperatures or major/minor-element proxies for “K” REE “P”). For your reference, Rapp & Draper (2018) include Na and K (and P) in their starting compositions performing very similar experiments, which may be more relevant for this study(?).
 13. **Supplementary Table 7.** It is stated that the MgO and SiO₂ values are assumed, but please clarify how the values were determined.
-

References Cited:

1. Milman-Barris, M.S. et al., (2008). Zoning of phosphorus in igneous olivine. *Contrib. Min. Petrol.*, 155, 739-765.
2. Prissel, T.C., and Gross, J. (2020). On the petrogenesis of lunar troctolites: New insights into cumulate mantle overturn and mantle exposures in impact basins. *Earth and Planetary Science Letters*, 551, 116531. <https://doi.org/10.1016/j.epsl.2020.116531>
3. Gross, J., et al. (2020). Geochemistry and Petrogenesis of Northwest Africa (NWA) 10401: A new type of the Mg-suite rocks. *J. Geophys. Res.: Planets*, e2019JE006225; <https://doi.org/10.1029/2019JE006225>
4. Yamamoto, S., et al., (2010). Possible mantle origin of olivine around lunar impact basins detected by SELENE. *Nat. Geosci.* 3, 533-536.
5. Melosh, H.J., et al., (2017). South Pole-Aitken basin ejecta reveal the Moon's upper mantle. *Geology*, 45, 1063-1066.
6. Moriarty, D.P., et al., (2020). Evidence for a stratified upper mantle preserved within the South Pole-Aitken Basin. *J. Geophys. Res. Planets*, 121, e2020JE006589. <https://doi.org/10.1029/2020JE006589>
7. Borg, L.E., et al., (2020). The formation and evolution of the Moon's crust inferred from the Sm-Nd isotopic systematics of highlands rocks. *Geochim. Cosmochim. Acta*, 290, 312-332.
8. Boukaré, X.-E., et al., (2018). Timing of mantle overturn during magma ocean solidification. *Earth Planet. Sci. Lett.*, 491, 216-225.
9. Li, H., et al., (2019). Lunar cumulate mantle overturn: a model constrained by ilmenite rheology. *JGR Planets*. <https://doi.org/10.1029/2018JE005905>.
10. Taylor, D.J., McKeegan, K.D., and Harrison, T.M. (2009). Lu-Hf zircon evidence for rapid lunar differentiation. *Earth and Planetary Science Letters*, 279, 157-164.
11. Barboni, M., Boehnke, P., Keller, B., et al. (2017). Early formation of the Moon 4.51 billion years ago. *Science Advances*, 3, doi: 10.1126/sciadv.1602365
12. Crow, C.A., McKeegan, K.D., and Moser, D.E. (2017). Coordinated U-Pb geochronology, trace element, Ti-in-zircon thermometry and microstructural analysis of Apollo zircons. *Geochimica et Cosmochimica Acta*, 202, 264-284.
13. Shearer, C.K., et al., (2015). Origin of the lunar highlands Mg-suite: An integrated petrology, geochemistry, chronology, and remote sensing perspective. *Am. Min.*, 100, 294–325.

14. Borg, L.E., et al. (2011). Evidence that the Moon is either young or did not have a global magma ocean. *Nature* 477, 70-72.
15. Borg, L.E., et al. (2017). Chronologic implications for slow cooling of troctolite 76535 and temporal relationships between the Mg-suite and the ferroan anorthosite suite. *Geochim. Cosmochim. Acta* 201, 377-391.
16. Pernet-Fisher, J.F., Deloule, E., and Joy, K.H. (2019). Evidence of chemical heterogeneity within lunar anorthosite parental magmas. *Geochim. Cosmochim. Acta*, 266, 109-130.
17. Xu, X., Hui, H., Chen, W., Huang, S., Neal, C.R., and Xu, X. (2020). Formation of lunar highlands anorthosites. *Earth and Planetary Science Letters*, 536, 116138.
18. Irvine, T.N. (1965). Chromian spinel as a petrogenetic indicator: Part 1. Theory. *Canadian J. Earth Sci.*, 2, 648-672.
19. Irvine, T.N. (1967). Chromian spinel as a petrogenetic indicator: Part 2. Petrologic applications. *Canadian J. Earth Sci.*, 4, 71-103.
20. Allan, J.F., et al. (1988). Cr-rich spinels as petrogenetic indicators; MORB-type lavas from the Lamont seamount chain, eastern Pacific. *Am. Min.*, 73, 741-753.
21. Dick, H.J.B., and Bullen, T. (1984). Chromian spinel as a petrogenetic indicator in abyssal and alpine-type peridotites and spatially associated lavas. *Contrib. Min. Petrol.*, 86, 54-76.
22. Kamenetsky, V.S., et al. (2001). Factors controlling chemistry of magmatic spinel: an empirical study of associated olivine, Cr-spinel and melt inclusions from primitive rocks. *J. Petrol.*, 42, 655-671.
23. Prissel, T.C., et al., (2016). Formation of the lunar highlands Mg-suite as told by spinel. *Am. Min.*, 101, 1624-1635.
24. Longhi, J., et al. (2010). The pattern of Ni and Co abundances in lunar olivines. *Geochim. Cosmochim. Acta*, 74, 784-798.
25. Albee, A.L., et al. (1974). Dunite from the lunar highlands: petrography, deformational history, Rb-Sr age. *Lunatic Asylum, Contrib. No. 2442*.
26. Dymek, R.F. et al. (1975). Comparative petrology of lunar cumulate rocks of possible primary origin: Dunite 72415, troctolite 76535, norite 78235, and anorthosite 62237. *Proc. 6th Lunar Sci. Conf.*, 301-341.
27. Lindstrom, M.M. et al. (1984). Magnesian anorthosites and associated troctolites and dunite in Apollo 14 breccias. *JGR*, 89, C41-C49.

28. Warren, P.H. (1986). Anorthosite assimilation and the origin of the Mg/Fe-related bimodality of pristine Moon rocks: Support for the magmasphere hypothesis. *JGR*, 91, D331-D343.
29. Ryder, G. (1991). Lunar ferroan anorthosites and mare basalt sources: The mixed connection. *GRL*, 18, 2065-2068.
30. Prissel, T.C. et al. (2014). Pink Moon: The petrogenesis of pink spinel anorthosites and implications concerning Mg-suite magmatism. *Earth and Planetary Science Letters*, 403, 144-156.
31. Morgan, Z. et al. (2006). An experimental study of anorthosite dissolution in lunar picritic magmas: implications for crustal assimilation processes. *Geochim. Cosmochim. Acta*, 70, 3477-3491.
32. Treiman, A.H., et al. (2019). Spinel-anorthosites on the Moon: impact melt origins suggested by enthalpy constraints. *Am. Mineral.*, 2014, 370-384.
33. Solomon, S.C., and Longhi, J. (1977). Magma oceanography: I. Thermal evolution. *Proc. Lunar Sci. Conf. 8th*, 583-599.
34. Toksoz, M.N., et al. (1978). Thermal evolutions of the terrestrial planets. *Moon and Planets*, 18, 281-320.
35. McCallum, I.S., and Schwartz, J.M. (2001). Lunar Mg-suite: thermobarometry and petrogenesis of parental magmas. *JGR*, 106, 27,969-27,983.
36. Rapp, J.F., and Draper, D.S. (2018). Fractional crystallization of the lunar magma ocean: updating the dominant paradigm. *Meteorit. Planet. Sci.* 53, 1432-1455.

Reviewer #2 (Remarks to the Author):

Review of "Fast magmatic cooling of the slowest-cooled Apollo sample" by Nelson et al.

Nelson and coworkers present very interesting and equally surprising (at least to me, as someone who has worked on this sample) observations of preserved magmatic zoning in the heavily annealed lunar crustal troctolite 76535. This sample is one of, if not the most heavily studied lunar sample and certainly one of the most important and informative samples of the Moon currently in human possession. Given that the rock contains virtually no chemical zoning of any type, these observations are unexpected and interesting. The authors combine these observations with 2D models of diffusive relaxation of the chemical zoning patterns to infer magmatic crystallization timescales. Their results clearly show a discrepancy between the timescales they infer and the subsolidus cooling timescales inferred from sample chronology and closure temperatures for radioisotope systems, the latter of which are more consistent with the thermally annealed nature of the rock. Separately, their results also imply that current knowledge of phosphorus diffusion in olivine needs to be revisited, although this is not stated explicitly. The authors then construct a quantitative thermal model for the size of the 76535 magma body and propose a thought-provoking qualitative model for the formation of Mg-suite suite (or at least troctolite) parental magmas that I believe constitutes a solid step forward.

These results will undoubtedly be of interest to the lunar science community. The observations are sound and the data collected are robust. The modeling efforts seem well-conceived and for the most part consider appropriate ranges for uncertain parameters. The paper is generally well-written, although it would benefit from a more careful read through. I have provided some comments below for the authors to consider and incorporate, however I see no barriers to publication and strongly recommend the manuscript to move forward in the publication process at Nature Communications, which is an appropriate journal for this work.

General Comments

I find the petrogenetic model envisioned in figure 5 to be interesting and thought provoking. Although my colleagues and I have long advocated for the hybridized source model for Mg-suite parental melts for a number of reasons, the physical mixing of deep dunitic cumulates with KREEP and crustal anorthosite has always seemed problematic to me. I think this model is a step toward shoring up that weakness. However, my concern is, how do the other Mg-suite lithologies fit into such a model? This paper is obviously about a troctolite, but I don't think one can or should treat their origins piecemeal. The norites and gabbronorites, especially those from Apollo 17, have textures, major element compositions, and REE abundances that indicate they may be direct fractional crystallization products of the same or similar parental magmas as troctolites such as 76535. In a magma chamber model this makes sense: the liquid evolves through fractionation and crystallizes troctolites, norites, and then gabbronorites. Troctolite parental melts are very olivine normative, so it seems that extensive fractional crystallization is required. In a reactive flow model, however, I do not see an obvious way to produce these other Mg-suite lithologies. So while I like the model proposed here and think it represents some creative thought regarding Mg-suite origins, as it is, I think it leads to a whack-a-mole problem: if troctolites formed in this fashion, how then do we make more evolved and just as important lithologies of the Mg-suite? I think the authors should consider this during their revisions and think about physical models that might be able to account for and explain their observations here while also leaving room for the broader Mg-suite.

More Specific Comments

Lines 18-19: Only FANs are crystallization products of the LMO, most lunar lithologies are not.

Lines 31 – 37: Carefully revisit the references in these lines. Although the refs cited did indeed invoke the processes they are associated with here, they were not the first papers to suggest those processes, as is stated.

Line 87: I believe there's a typo in the subscript of D-Plag. Should be Ca-Na, not Ca-Mg.

Line 95: Delete 'other', P is not tetravalent.

Line 103: Please define what functions and coefficients are being referred to. Cooling pathways for functions or did the authors use different equations? Does 'coefficients' refer to diffusion coefficients or the fit coefficients (a and b)? If 'coefficients' refers to varying possible diffusion coefficients, insert 'diffusion' here so as to be specific.

Lines 114 – 115: References for previous estimates?

Line 116: This probably does not need to be its own section, as the recrystallization section is only a paragraph long. More broadly, I think the authors can rethink their sectioning of the paper to reduce the number, this being one example.

Lines 198 – 204: Give the amount of annealing and recrystallization that has occurred in 76535, I am unsure how any of the plagioclase grains in the rock meet criteria (a) here. Please provide more info.

Line 342: Typo? Should read 'equation 3' instead of 4, I believe.

Lines 377 – 381: I'm not sure I agree that we have no independent constraints on crustal temperatures, or that the crust was as cool as assumed here. Since emplacement of the Mg-suite pretty much has to occur essentially contemporaneously with the end of the LMO, the crust, especially the lower crust, can't be much below its solidus. Additionally, work by Meyer et al. (2010) *Icarus* has modeled the temperature evolution of the crust (and the effects of tidal heating on it), and in my 2020 *Nature Geoscience* paper, Matthieu Laneuville's modeling of heat production from KREEP shows just how much heat is produced from radioactive decay, which will undoubtedly keep the crust quite warm. The two-pyroxene temperatures from McCallum and Schwartz (2001) and my 2012 paper both show 76535 sitting at 800 – 900 C when (presumably) it was excavated from the lower crust, so those temps, combined with the very slow cooling rate from Borg et al. suggest the crust was warmer than assumed here. I think it's worth checking out the references here, especially the Meyer paper, and consider a thermal model with a hotter crust.

References: My 2020 *Nature Geoscience* paper is in the reference list twice (8 and 38). Although I'd be happy to have this count as two citations instead of one, it won't work that way, so better to just cite it once.

Steve Elardo
University of Florida
5/20/2021

Reviewer #3 (Remarks to the Author):

This manuscript reports important significant observations on lunar sample 76535, the much-studied and photogenic archetype of the lunar Mg-suite plutonic rocks. The manuscript should be published in nearly its present form. I found it to be well-written, well-organized, and compelling.

My comments below should be dealt with easily – as mostly they are addenda to the arguments. One will require a bit of reconsideration or presentation of data not already in the manuscript.

General Comments

1. The authors' discussion of zoning in the 76535 olivine and modeling thereof (page 3, Fig. 1) needs to include information on the orientation of the P-enriched lamellae relative to the thin section plane. This is important, because this effect of thin section geometry can lead to uncertainties of an order of magnitude in diffusion timescales because of geometry and diffusion anisotropy (Shea et al. 2015). This uncertainty affects diffusion modeling of the widths of the P-rich lamellae, so that lamellae at lower angles with respect to the section plane should appear to widen faster than those at high angles – this difference is not apparent in the modeled profiles (Figs. 1, 3).

It would be helpful if the authors could define the crystallographic planes on which the P-rich lamellae lie – perhaps by analogy with experimentally produced lamellae or well-characterized natural ones. Alternatively, perhaps the authors could determine the orientations of

the olivine grains by EBSD. Then, they could calculate the angles between the section plane and the lamellae's planes.

2. The authors explain the absence of lamellae in most olivines, and most parts of the olivines with lamellae, by dissolution/reprecipitation during a later event of partial melting initiated by infiltration of basaltic melt. I've no argument with this, but homogenous olivine can form in other ways. Most specifically for the olivine of Figure 1f-1i, formation and crystallization of a melt inclusion can erase P zoning. For example, Goodrich et al. (2013) show strong lamellar P-zoning in olivine in the Nakhla martian meteorite, and that melt inclusions in the olivine are surrounded by areas of constant low P (her Figs. 2, 3). These are (or can be) interpreted as cognate olivine, grown from the melt inclusion at leisure as the rock cooled. Another possible mechanism for erasing P-zoning is migration and coalescence of melt inclusions. In a rapidly grown dendritic olivine, trapped melt will lie in elongated discs or rods among the dendrites, and it can coalesce (by solution & reprecipitation) to single larger inclusions to minimize surface area/energy.

3. Although lamellar P-zoning is recognized wherever basalts are found, the authors might add references for such zoning in other lunar samples. For example, Elardo and Shearer (2014) for a lunar meteorite basalt and Demidova et al. (2018) for L20 soil grains. Similarly, the authors could point to lunar materials inferred to have cooled slowly, xenoliths in basalt, that do not show zoning in P (Shearer et al. 2015).

4. I'm unconvinced by the formation scenario section of the manuscript in two respects – the second scenario seems unlikely, and a third (uncited) scenario seems possible. In the second scenario (lines 167 ff), the authors would have "... mafic cumulate diapirs flow[ing] into the lower crust...". This seems unlikely to me as mafic cumulate diapirs would have been dense, and

the lunar crust, being composed primarily of plagioclase, would have been less dense. Hence, there would be no driving force to make the diapirs rise into the crust – they would stall at the crust-mantle boundary. The authors should explain their model further.

A third scenario seems possible, that 76535 formed from an impact melt. One can readily imagine an impact large enough to melt and mix material from the crust and uppermost mantle, yielding a melt of the proper composition to parent 76535. That melt would be superheated, and cool quickly down below the olivine-liquidus, allowing rapid growth of olivine and formation of P-rich lamellae. Cooling would slow down considerably thereafter, as the melt would have been emplaced into (or onto) crust that had been strongly heated by the impact event. I've no numbers to go with this scenario, but it seems plausible *prima facie*.

5. The analytical conditions for plagioclase are a concern. I usually analyze plagioclase at much milder conditions, 10-30 nanoamps beam current, because Na can be burnt out of the plagioclase. I wonder if the plagioclase grains without apparent Na zoning were analyzed at high beam currents, like the 100 nA.

Smaller comments

1. The title is confusing, how can it have cooled fast and slow simultaneously? To help the reader, I recommend revising the title to something like “Rapid magmatic cooling of lunar troctolite 76535 – Evidence against a plutonic origin for this Mg-suite ‘plutonite.’ ”
2. References 8 and 38 are the same.
3. Several picky comments through the text, mostly unimportant to the sense of the paper.

REFERENCES.

- Goodrich, C. A., Treiman, A. H., Filiberto, J., Gross, J., & Jercinovic, M. (2013). K₂O-rich trapped melt in olivine in the Nakhla meteorite: Implications for petrogenesis of nakhlites and evolution of the Martian mantle. *Meteoritics & Planetary Science*, *48*(12), 2371-2405.
- Demidova, S. I., Ntaflos, T., & Brandstätter, F. (2018). P-bearing olivines from the “Luna-20” soil samples, their sources and possible phosphorus substitution mechanisms in lunar olivine. *Petrology*, *26*(3), 314-327.
- Elardo, S. M., & Shearer Jr, C. K. (2014). Magma chamber dynamics recorded by oscillatory zoning in pyroxene and olivine phenocrysts in basaltic lunar meteorite Northwest Africa 032. *American Mineralogist*, *99*(2-3), 355-368.
- Shea, T., Costa, F., Krimer, D., & Hammer, J. E. (2015). Accuracy of timescales retrieved from diffusion modeling in olivine: A 3D perspective. *American Mineralogist*, *100*(10), 2026-2042.
- Shearer, C. K., Burger, P. V., Bell, A. S., Guan, Y., & Neal, C. R. (2015). Exploring the Moon's surface for remnants of the lunar mantle 1. Dunite xenoliths in mare basalts. A crustal or mantle origin? *Meteoritics & Planetary Science*, *50*(8), 1449-1467.

Response to reviewers:

We thank the reviewers for providing exceptionally helpful and scholarly reviews. Below, we include the original reviewer texts followed by our responses (in blue), including explicit descriptions of the changes we made to the manuscript during revision.

Reviewer 1:

Nelson et al., present chemical analysis of and diffusion modeling for Apollo troctolite 76535 in order to further constrain petrogenesis of the enigmatic highlands Mg-suite. The authors find Phosphorus (P) heterogeneities in olivine, which are consistent with terrestrial analogs and indicative of rapid cooling from magmatic temperatures. Modeling results and analog comparisons suggest that sample 76535 cooled at a rate >2 orders of magnitude faster than previous estimates. The authors then synthesize their findings with some previous models of origin and discuss qualitative implications concerning the existing package of petrogenetic models.

In general, I find that the authors have conducted a detailed chemical analysis of 76535 and extract new data on one of the most well-studied samples in the Apollo collection. The analysis and diffusion modeling presents significant compositional, temporal, and thermal constraints for the origin of the Mg-suite. Because the formation of Mg-suite is intimately tied to constraining the guiding paradigm of lunar differentiation (lunar magma ocean), the topic is well-suited for the readership of Nature Communications.

However, I find that the manuscript undersells its important compositional, temporal, and thermal constraints presented and as a result, overreaches with respect to some of its qualitative comparisons. Unfortunately, a rapid shift to the qualitative comparisons raised a number of quantitative questions regarding the proposed model, and also how the new results factor into previous conclusions. In providing more detailed analysis of their findings, the discussion should follow a systematic evaluation of the current models. Here, each model of origin should be filtered through the same set of chemical, temporal, and thermal constraints defined by their measurements and modeling in determining which hypotheses require further attention or which hypotheses can potentially be ruled out on this basis.

Below, I've outlined five considerable considerations related to the compositional, temporal, and thermal constraints presented. These are followed by some relatively minor comments related to section/line aspects in the manuscript. After careful consideration, I recommend that the paper be returned to the authors in order to address these comments for revision. I have selected topics of discussion that I believe will strengthen the results by framing into a more quantitative and comparative context, help broaden the current scope, and present the reader with important and relevant findings of fact for future considerations and further scrutiny of the present data set.

We acknowledge the oversight of Prissel and Gross (2020), which is an important new

contribution to Mg-suite petrogenesis that we now discuss and reference in the revised manuscript.

1. The results motivate reexamination of other olivine-bearing Mg-suite samples in order to identify their cooling rates based on the methods and P-analyses presented. Thus, the results should be incorporated into a more comprehensive cooling history for Mg-suite rocks in general (or to understand any variances among location and landing sites). This motivation is not directly/explicitly expressed. I find this to be a primary topic and implication of the study in and of itself, and the manuscript should not shy away. I find that it is definitely worth reviewing/noting up front since it **a)** forwards a method for adding value to the existing Apollo collection in addition to **b)** providing a means to further scrutinize the conclusions of the manuscript.

We strongly agree

What we did to address this comment: we added a paragraph in the discussion explicitly calling for new investigations into other Mg-suite samples (Lines 251-264).

- a. Importantly, the manuscript does not revisit the previous models that proposed much slower cooling histories. How do the new and faster cooling rates support/contradict the previous findings of (e.g.,) exsolution and ordering in pyroxenes? Can the exsolution lamellae and ordering be explained with the faster cooling rates, or do they still require the slower cooling rates? If the latter, how are these two issues reconciled?

We agree that this is an important discussion to emphasize. In this study, we do not actually contradict any prior measurements of cooling rates, but rather question the interpretations of those measurements. For this example, cooling rates obtained by McCallum et al., 2006 were determined only around the closure temperature for Mg/Fe, or 500-530 degrees Celsius. Different geospeedometers constrain different portions of the cooling history. Thus, our results and these previous findings work together to provide one coherent cooling path of the sample, where our estimate pertains to the high temperature portion of this path.

What we did to address this comment: We added a paragraph to better emphasize this notion and provide clarity to the reader (Lines 147-152).

2. All throughout, the manuscript appears to favor a hybridized source model which requires that primary mantle cumulates mixed with crust + KREEP prior to then partially melting to form the KREEP-bearing cotectic Mg-suite parental melt. After careful review, some of the data presented in the current manuscript appears to contradict the present conclusions. Further, the manuscript should provide more context to the issue of

petrogenesis such as KREEP-poor meteorites, previous experimental work testing the hybridized source model, and orbital remote sensing. It is for these reasons a more critical analysis is necessary in order to justify the present conclusions or incorporate important amendments during revisions.

- a. The maximum reported abundance of P₂O₅ as measured by the author's 5-spectrometer method is ~ 0.1 wt.%, and with a minimum of 0.006 wt.% P₂O₅ (noting that this range is defined by a single profile measured on a single crystal of olivine). Building upon your terrestrial analog (and to my knowledge), the present 76535 P-values are consistent with P-measurements in olivine for many mantle-derived terrestrial samples (0.006 – 0.2 wt.% P₂O₅; Milman-Barris et al., 2008). The P-in-olivine correlations between 76535 and mantle-derived terrestrial samples seem to simply suggest that 76535 was also derived from mantle-melting (e.g., Longhi, 1981; Prissel and Gross, 2020), and that there is no need for a complex hybridized source. Can you demonstrate that similar P-abundances in olivine are expected in both the terrestrial mantle-derived samples and Mg-suite troctolites derived from a hybridized KREEP-rich source?

This is an interesting question. Shea et al., 2019 argued that P is distributed as though nearly partitionless between olivine and melt at relatively high growth rates (which we infer to be the case here based on the skeletal P blueprint of 76535 olivine), implying that the melt would also be near 0.1 wt% P₂O₅. Should the olivine crystallize from ur-KREEP, the max P content would be between 3-4 wt% P₂O₅ in the melt, and thus the olivine in 76535 (Warren and Watson, 1979). However, this scenario is unlikely as ur-KREEP is expected to be a minor component in the parent magma. Given that Gross et al., 2020 find the bulk rock P₂O₅ of NWA 10401 (A Mg-suite like, KREEP-poor meteorite) to be around 0.01 wt%, an order of magnitude lower than the parent melt of 76535. The P concentrations in these heterogeneities would imply that it is quite possible that the KREEP signature is present before olivine crystallizes in Mg-suite samples. However, as the reviewer stated, P-enriched olivine can come from a variety of sources, certainly not all KREEP-enriched (e.g. Milman-Barris et al., 2008; Welsch et al., 2014). Thus, this line of reasoning cannot be applied to argue for or against a pre-crystallization KREEP enrichment of the parental melt.

What we did to address this comment: Clarification of this topic is included in the edits to our discussion (Lines 231-250) of how a reactive melt model does not preclude a non-hybridized source.

- b. How does the proposed hybridized source model satisfy KREEP-poor and KREEP-rich magnesian-suite-like meteorite clasts (e.g., Gross et al., 2020) originating from source localities outside of the PKT?

The reviewer points out that we do not mention a category of candidate magnesian suite samples from the meteorite collection that lack the KREEP signature that is present in Apollo Mg-suite samples. We agree that it is important for our model to explain *all* Mg-suite samples.

What we did to address this comment: We included discussion on this subject (Lines 227-230)

- c. Another important constraint from orbital remote sensing: the hybridized source model suggests olivine cumulates are transported to the base of the lunar crust. But orbital remote sensing indicates that olivine mantle exposures in large impact basins are rare (e.g., Yamamoto et al., 2010; Melosh et al., 2017; Moriarty et al., 2020).

Prissel and Gross (2020) circumvented this issue by suggesting that Mg-suite mantle-melts may have been extracted in a similar manner to picritic glasses. How does the proposed hybridized source model reconcile the paucity of olivine-mantle exposures in the remote sensing database?

The reviewer brings up an ongoing argument that is part of Mg-suite petrogenesis. We are not completely convinced by this argument. Since the findings reported in the papers above were published, the Chang'e-4 rover has reported the presence of forsteritic olivine in equal proportions to Mg-rich pyroxene (Gou et al., 2020 Icarus V345) in the SPA basin. The physical modelling done by Melosh et al. (2017) suggests the rocks at this site are largely a sample of the lunar mantle. A mixture of olivine and LCP in the upper mantle appears most likely, which is fully in line with the hybridized source model. However, our interpretations are not fatally inconsistent with either model.

What we did to address this comment: We clarified that our model does not necessarily require a hybridized source (Lines 231-250).

3. The manuscript presents diffusion modeling which constrains the cooling timescales of Mg-suite to < 27My (or < 20My based on olivine alone). If so, this severely limits the expected or anticipated range of crystallization ages of Mg-suite samples from the isotopic system of interest.
 - a. Borg et al., (2020) report the Mg-suite on a whole crystallized ~4340 +/- 9Ma. This tightly constrained formation age of Mg-suite is not necessarily expected if the cooling timescales for some Mg-suite samples in the crust are capable of exceeding 100My time prior to isotopic closure. I think this is an important result that adds relevant discussion to temporal issues of Mg-suite in addition to providing further context to the more confounding issue in lunar science – the near-contemporaneous relationship between LMO-products and Mg-suite.

Agreed.

What we did to address this comment: We included a short discussion on the contemporaneous crystallization of the Mg-suite (Lines 260-264), and the potential to utilize our findings to catalyze future research in this field.

- b. Geodynamic models suggest the onset of mantle overturn can occur near-contemporaneously with LMO solidification (e.g., Boukaré et al., 2018; Li et al., 2019), and so Mg-suite models directly tied to mantle overturn appear to satisfy this issue. The decompression-melting model proposed by Prissel and Gross (2020) demonstrated that partial melting of primary LMO cumulates as a direct result of mantle overturn can produce successful KREEP-free Mg-suite primary melts. The hybridized source hypothesis is not as direct or straight forward, and it is unclear how quickly mantle cumulates can mix in the precise proportions to produce successful Mg-suite melts as presently defined (it is only assumed to be instantaneous relative to the onset of mantle overturn, but this is not quantified – Shearer et al., 2015). The present manuscript is not necessarily equipped to address the unquantified time constraints associated with the hybridized source hypothesis, but these are important aspects that deserve attention in the discussion.

We do not see the hybridized source model as more (or less) complicated than the model put forward by Prissel and Gross (2020), but we agree that the discussion of this topic in the original manuscript gave the impression of supporting only the hybridized source model, which it does not.

What we did to address this comment: we rewrote our discussion to broaden our model to permit parent melts from many different sources (Lines 231-250).

- c. Coupled U-Pb and Lu-Hf studies suggest rapid LMO solidification and older LMO model ages (e.g., Taylor et al., 2009; Barboni et al., 2017; Crow et al., 2017). In particular, can your interpretations of rapid Mg-suite crystallization be reconciled with $^{207}\text{Pb}/^{206}\text{Pb}$ crystallization ages of lunar zircon and Lu-Hf model ages of LMO solidification at ~4.51 Ga (Barboni et al., 2017), followed by mantle-overturn- induced Mg-suite ~200 My later at 4.34 Ga?

Agreed. Our model cannot explicitly discuss anything that occurred prior to the onset of Mg-suite crystallization. However, we can address how our results do fit into the larger picture of lunar formation, crystallization, and overturn.

What we did to address this comment: We added a paragraph discussing this topic (Lines 265-278).

- d. Some further considerations: see also the slow-cooling magma ocean model recently proposed by Maurice et al. (2020) that is consistent with an initial LMO solidification age of 4425 ± 25 Ma, which predates the formation of Mg-suite and suggests that the younger formation age of 60025 (4359 ± 9 Ma; Borg et al., 2020) is unrepresentative of the earliest formed crust and instead may represent lunar crustforming at the final stages of a slow-cooling magma ocean. In this way, Mg-suite crystallization ages do not necessarily pre-date the formation of all lunar crust.

The reviewer makes a good point, and we had considered this line of reasoning. Following the discussion in Borg et al. (2017), we decided to retain discussion of the timing of crystallization for 60025 due to it having the most precise crystallization age in the Ferroan anorthosite suit (Borg et al. 2011), posing the paradox discussed in the paper, which is only made worse by the even younger date of FAS 62237 (Sio et al., 2020) We resolve this paradox by demonstrating rapid magmatic cooling of troctolite 76535, the very Mg-suite sample that generated this paradox.

- e. The authors may find it also relevant to add discussion on the complications associated with interpreting the age data itself including low-closure temperatures for the Sm-Nd isotopic system and thus, potential isotopic resetting events such as mantle overturn (or Mg-suite magmatism itself), KREEP metasomatism (if operative), giant basin forming impacts, radioactive heating within the Procellarum KREEP Terrane, or a possible combination of these (e.g., Borg et al., 2011, 2017; Barboni et al., 2017; Pernet-Fisher et al., 2019; Xu et al., 2020).

What we did to address this comment: Added a paragraph to the discussion (lines 261-278) and an additional discussion on the topic of thermal resetting (that diffusion will record any reheating events equally, and thus late reheating would only further shorten the possible crystallization timescale) (lines 143-146).

4. The paper presents a nice terrestrial analog for discussion purposes as it relates to a cumulate origin. However, the terrestrial analog contains Cr-spinel and this is an important petrogenetic indicator (e.g., Irvine 1965, 1967; Dick and Bullen 1984; Allan et al. 1988; Kamenetsky et al. 2001), particularly as it pertains to the Mg-suite (Prissel et al., 2016).
 - a. Prissel et al. (2016) experimentally tested the hybridized source model – summarizing: partial melting of the hybridized source yields troctolitic parent melts (olivine+plagioclase saturated; Longhi et al., 2010) as modeled here

(Supplementary Table 1) with > 15 wt.% Al₂O₃. However, Prissel et al., (2016) experimentally demonstrate why troctolitic/cotectic magmas are potentially problematic as Mg-suite parental melts. The reason being is that troctolitic parent melts stabilize pink spinel owing to the high MgO and Al₂O₃ starting concentrations. Prissel and Gross (2020) point out that pink spinel troctolites represent < 2 wt.% of all lunar troctolites, whereas common lunar troctolites (> 98wt.% of all lunar troctolites) contain magmatic Cr-spinel or evidence of magmatic chromite (or no pink spinel; e.g., Albee et al. 1974; Dymek et al. 1975; Lindstrom et al. 1984; Shearer et al. 2015).

- b. Cr-spinel is of course ubiquitous among mantle-derived olivine-rich samples (see also the terrestrial troctolite discussed from the Atlantis Massif). Drawing from earlier suggestions (Warren, 1986; Ryder, 1991), and results from melt-anorthite reaction experiments (Prissel et al., 2014), Prissel and Gross (2020) concluded that minor amounts of pink spinel can be explained from minor amounts of assimilation of the lunar crust. The minor amount of assimilation required to produce minor amounts of pink spinel is consistent with anorthosite assimilation being a minor aspect (and likely not always operative) of petrogenesis as defined by experimental petrology and thermodynamic modeling (e.g., Morgan et al., 2006; Treiman et al., 2019). If the hybridized source model is the predominant mode of origin and yield troctolitic (or cotectic) melts, why then are pink spinel troctolites so extremely rare (Prissel et al., 2016)? Or put another way, why is the occurrence of magmatic Cr-spinel (or evidence thereof and lack of pink spinel) in > 98 wt.% of all lunar troctolite samples *not* indicative of a mantle origin (Borg et al., 2020; Prissel and Gross, 2020)?

The reviewer raises important considerations involving the mineralogy of the spinel phase present in various lunar Mg-suite samples, specifically the implications for source rock(s) indicated when the rock contains the Al-rich spinel (a.k.a. “pink” spinel) versus a Cr-rich spinel. His summary of the constraints afforded by spinel mineralogy concludes with a prompt to comment on the suitability of the hybrid source and/or to explain why the decompression melting model is *not* suitable, applying the constraint of spinel mineralogy. We concur that the original submitted version of our manuscript did not include discussion of the decompression melting model. Insofar as our results do not provide new insights from or about spinel, we modified our discussion to include the physical configuration of liquid and solid phases consistent with the decompression-melting model.

What we did to address this comment: We modified the discussion to incorporate, explicitly, the leading melting models for the troctolite source: hybrid-source and decompression mantle melting. We included a statement acknowledging the relative simplicity of forming troctolite by reactive infiltration involving a partial melt of the mantle (as opposed to a hybrid source) (Lines 231-239).

5. The manuscript offers an example result if the crust were cooler than their assumed 650°C input for modeling, and suggests no independent constraints are available. However, independent modeling of the thermal evolution of the lunar crust suggests at the time of Mg-suite magmatic emplacement the temperatures of the lunar crust could be hotter, near 800°C (e.g., Solomon and Longhi 1977; Toksoz et al., 1978; Warren, 1986). Moreover, Warren (1986) notes that models of early lunar thermal evolution did not consider the insulating effects of a thick megaregolith, which could've further increased local temperatures by ~100°C. In a figure of choice, the manuscript would be strengthened by quantitatively evaluating the effects of a cooler and hotter crust.

Multiple reviewers made this point: We provided data for an unrealistically low temperature of crust, and didn't adequately show that this would lead to the largest possible magma chamber.

What we did to address this comment: We ran the models at multiple different temperatures and provided results in Supplementary Figure 8.

-
6. **Title:** The title is nondescript relative to its content. The authors should reconsider working in some common and descriptive search terms of their liking such as 'lunar troctolite,' 'phosphorus,' 'diffusion,' and/or the sample number 76535.

What we did to address this comment: We appreciate the reviewer's suggestion, and have changed the title to "Chemical heterogeneities Reveal Early Rapid Cooling of Apollo Troctolite 76535"

7. **Abstract:** I recommend adding some quantification/results to the abstract. For instance, the element phosphorus is not even mentioned, but it is the cornerstone of this work. What are the cooling rates, and how different are they than previously estimated? Adding this information from the present manuscript will provide the glancing reader valuable and enticing information to dig deeper.
 - a. The authors may want to also reconsider the implications stated in the abstract after addressing the comments below.

What we did to address this comment: Restructured the abstract, adding in the suggested changes, and removing mention of the source region in favor of a more nuanced discussion in the main text of the article

8. **73:** In general, try to avoid using the qualifier "all" when citing only a fraction of all studies.

What we did to address this comment: We removed "all"

- 9. 79-80, 118-119:** The results would be strengthened by demonstrating in a figure that the oscillatory P-zoning you measured is independent of normal major element zoning (Mg/Fe) over the same region. And then describe in some more detail why previous work/experiments suggest rapid crystallization vs. other hypotheses such as compositional-fluxes during magma mingling.

What we did to address this comment: To address the first part of the comment, we added Supplementary Figure 6. As for the second portion, we added a reference to Milman-Barris (2008) that an interested audience could go to for more details. However, we do not think it fits the flow of the article to explicitly walk through this, as it is widely accepted (Milman-Barris et al. 2008; Welsch et al. 2013, 2014; Shea et al. 2015; de Maisonneuve et al. 2016; Manzini et al. 2017; Xing et al. 2017; Baziotis et al. 2017; Ersoy et al. 2019).

- 10. 85-100:** Previous studies have identified sub-solidus re-equilibration between olivine-spinel as an operative mechanism for Mg-suite samples (e.g., McCallum and Schwartz, 2001; Prissel et al., 2016). In particular, Prissel et al. (2016) demonstrate that pink spinel troctolites appear in olivine-spinel magmatic equilibrium, whereas Cr-spinel bearing Mg-suite samples appear to have experienced some Fe/Mg sub-solidus re-equilibration. These results were used to suggest pink spinel troctolites (cooled rapidly at the chill margins of a magmatic intrusion) whereas common lunar troctolites cooled more slowly at sub-solidus temperatures. Although the current study focuses on magmatic cooling, it might be worth mentioning to what extent (if any) we should expect P-heterogeneities that were locked in during rapid magmatic cooling to be later effected during sub-solidus heating events.

What we did to address this comment: We added a brief discussion on the effect of subsequent reheating towards the end of the modeling section (143-146).

- 11. 303-309:** It is stated that the parent composition in Table 1 was, in part, calculated based on the melt inclusion found in a lunar norite. Can you justify why using an evolved melt inclusion in equilibrium with orthopyroxene and plagioclase is appropriate for estimating a parental melt that first produced dunite → troctolite prior to norite assemblages? Why is this composition more appropriate than testing previous estimates of the Mg-suite parental melt (e.g., Longhi, et al., 2010; Prissel et al., 2016; Prissel and Gross, 2020)? Please provide more clarification as to how you determined the starting P and K abundances, or why they are assumed.

We chose the Mg-suite norite melt inclusion as a starting point for our melt as it is the only direct sample of the Mg-suite parent melt that we know of. Note that this magma composition is only used to determine the mineral-in temperatures that we use to obtain the initial temperatures for the diffusion modeling.

12. Figure 4: Lin et al⁴⁷ is referenced but not listed in the bibliography. If the authors are referring to any one of the LMO experimental studies from Yanhao Lin, please be advised that this series of papers did not use Na or K (or P) in the starting compositions (i.e., as it pertains to FAS compositions and liquidus temperatures or major/minor-element proxies for “K” REE “P”). For your reference, Rapp & Draper (2018) include Na and K (and P) in their starting compositions performing very similar experiments, which may be more relevant for this study(?).

We thank the reviewer for the Rapp and Draper citation.

What we did to address this comment: We updated Figure 4 to work with this superior source, though it only changed the plot by 10 degrees.

13. Supplementary Table 7. It is stated that the MgO and SiO₂ values are assumed, but please clarify how the values were determined.

Excellent point! We assumed pure forsterite to determine the phosphorus content in the olivine. However, we agree that this is confusing.

What we did to address this comment: We swapped a pure forsterite assumption for major element composition for those determined in our analysis of the olivine in slide ,52. The change in composition does not impact model times, as diffusion chronometry involving phosphorus is dependent only on relative P concentration. However, this has been updated and made more clear (Supplementary Table 7).

References Cited:

1. Milman-Barris, M.S. et al., (2008). Zoning of phosphorus in igneous olivine. *Contrib. Min. Petrol.*, 155, 739-765.
2. Prissel, T.C., and Gross, J. (2020). On the petrogenesis of lunar troctolites: New insights into cumulate mantle overturn and mantle exposures in impact basins. *Earth and Planetary Science Letters*, 551, 116531. <https://doi.org/10.1016/j.epsl.2020.116531>
3. Gross, J., et al. (2020). Geochemistry and Petrogenesis of Northwest Africa (NWA) 10401: A new type of the Mg-suite rocks. *J. Geophys. Res.: Planets*, e2019JE006225; <https://doi.org/10.1029/2019JE006225>
4. Yamamoto, S., et al., (2010). Possible mantle origin of olivine around lunar impact basins detected by SELENE. *Nat. Geosci.* 3, 533-536.
5. Melosh, H.J., et al., (2017). South Pole-Aitken basin ejecta reveal the Moon's upper mantle. *Geology*, 45, 1063-1066.
6. Moriarty, D.P., et al., (2020). Evidence for a stratified upper mantle preserved within the South Pole-Aitken Basin. *J. Geophys. Res. Planets*, 121, e2020JE006589. <https://doi.org/10.1029/2020JE006589>
7. Borg, L.E., et al., (2020). The formation and evolution of the Moon's crust inferred from the Sm-Nd isotopic systematics of highlands rocks. *Geochim. Cosmochim. Acta*, 290, 312-332.
8. Boukaré, X.-E., et al., (2018). Timing of mantle overturn during magma ocean solidification. *Earth Planet. Sci. Lett.*, 491, 216-225.
9. Li, H., et al., (2019). Lunar cumulate mantle overturn: a model constrained by ilmenite rheology. *JGR Planets*. <https://doi.org/10.1029/2018JE005905>.
10. Taylor, D.J., McKeegan, K.D., and Harrison, T.M. (2009). Lu-Hf zircon evidence for rapid lunar differentiation. *Earth and Planetary Science Letters*, 279, 157-164.
11. Barboni, M., Boehnke, P., Keller, B., et al. (2017). Early formation of the Moon 4.51 billion years ago. *Science Advances*, 3, doi: 10.1126/sciadv.1602365
12. Crow, C.A., McKeegan, K.D., and Moser, D.E. (2017). Coordinated U-Pb geochronology, trace element, Ti-in-zircon thermometry and microstructural analysis of Apollo zircons. *Geochimica et Cosmochimica Acta*, 202, 264-284.

13. Shearer, C.K., et al., (2015). Origin of the lunar highlands Mg-suite: An integrated petrology, geochemistry, chronology, and remote sensing perspective. *Am. Min.*, 100, 294–325.
14. Borg, L.E., et al. (2011). Evidence that the Moon is either young or did not have a global magma ocean. *Nature* 477, 70-72.
15. Borg, L.E., et al. (2017). Chronologic implications for slow cooling of troctolite 76535 and temporal relationships between the Mg-suite and the ferroan anorthosite suite. *Geochim. Cosmochim. Acta* 201, 377-391.
16. Pernet-Fisher, J.F., Deloule, E., and Joy, K.H. (2019). Evidence of chemical heterogeneity within lunar anorthosite parental magmas. *Geochim. Cosmochim. Acta*, 266, 109-130.
17. Xu, X., Hui, H., Chen, W., Huang, S., Neal, C.R., and Xu, X. (2020). Formation of lunar highlands anorthosites. *Earth and Planetary Science Letters*, 536, 116138.
18. Irvine, T.N. (1965). Chromian spinel as a petrogenetic indicator: Part 1. Theory. *Canadian J. Earth Sci.*, 2, 648-672.
19. Irvine, T.N. (1967). Chromian spinel as a petrogenetic indicator: Part 2. Petrologic applications. *Canadian J. Earth Sci.*, 4, 71-103.
20. Allan, J.F., et al. (1988). Cr-rich spinels as petrogenetic indicators; MORB-type lavas from the Lamont seamount chain, eastern Pacific. *Am. Min.*, 73, 741-753.
21. Dick, H.J.B., and Bullen, T. (1984). Chromian spinel as a petrogenetic indicator in abyssal and alpine-type peridotites and spatially associated lavas. *Contrib. Min. Petrol.*, 86, 54-76.
22. Kamenetsky, V.S., et al. (2001). Factors controlling chemistry of magmatic spinel: an empirical study of associated olivine, Cr-spinel and melt inclusions from primitive rocks. *J. Petrol.*, 42, 655-671.
23. Prissel, T.C., et al., (2016). Formation of the lunar highlands Mg-suite as told by spinel. *Am. Min.*, 101, 1624-1635.
24. Longhi, J., et al. (2010). The pattern of Ni and Co abundances in lunar olivines. *Geochim. Cosmochim. Acta*, 74, 784-798.
25. Albee, A.L., et al. (1974). Dunite from the lunar highlands: petrography, deformational history, Rb-Sr age. *Lunatic Asylum, Contrib. No. 2442*.
26. Dymek, R.F. et al. (1975). Comparative petrology of lunar cumulate rocks of possible

- primary origin: Dunite 72415, troctolite 76535, norite 78235, and anorthosite 62237. Proc. 6th Lunar Sci. Conf., 301-341.
27. Lindstrom, M.M. et al. (1984). Magnesian anorthosites and associated troctolites and dunite in Apollo 14 breccias. JGR, 89, C41-C49.
 28. Warren, P.H. (1986). Anorthosite assimilation and the origin of the Mg/Fe-related bimodality of pristine Moon rocks: Support for the magmasphere hypothesis. JGR, 91, D331-D343.
 29. Ryder, G. (1991). Lunar ferroan anorthosites and mare basalt sources: The mixed connection. GRL, 18, 2065-2068.
 30. Prissel, T.C. et al. (2014). Pink Moon: The petrogenesis of pink spinel anorthosites and implications concerning Mg-suite magmatism. Earth and Planetary Science Letters, 403, 144- 156.
 31. Morgan, Z. et al. (2006). An experimental study of anorthosite dissolution in lunar picritic magmas: implications for crustal assimilation processes. Geochim. Cosmochim. Acta, 70, 3477-3491.
 32. Treiman, A.H., et al. (2019). Spinel-anorthosites on the Moon: impact melt origins suggested by enthalpy constraints. Am. Mineral., 2014, 370-384.
 33. Solomon, S.C., and Longhi, J. (1977). Magma oceanography: I. Thermal evolution. Proc. Lunar Sci. Conf. 8th, 583-599.
 34. Toksoz, M.N., et al. (1978). Thermal evolutions of the terrestrial planets. Moon and Planets, 18, 281-320.
 35. McCallum, I.S., and Schwartz, J.M. (2001). Lunar Mg-suite: thermobarometry and petrogenesis of parental magmas. JGR, 106, 27,969-27,983.
 36. Rapp, J.F., and Draper, D.S. (2018). Fractional crystallization of the lunar magma ocean: updating the dominant paradigm. Meteorit. Planet. Sci. 53, 1432-1455.

Reviewer 2:

Nelson and coworkers present very interesting and equally surprising (at least to me, as someone who has worked on this sample) observations of preserved magmatic zoning in the heavily annealed lunar crustal troctolite 76535. This sample is one of, if not the most heavily studied lunar sample and certainly one of the most important and informative samples of the Moon currently in human possession. Given that the rock contains virtually no chemical zoning of any type, these observations are unexpected and interesting. The authors combine these observations with 2D models of diffusive relaxation of the chemical zoning patterns to infer magmatic crystallization timescales. Their results clearly show a discrepancy between the timescales they infer and the subsolidus cooling timescales inferred from sample chronology and closure temperatures for radioisotope systems, the latter of which are more consistent with the thermally annealed nature of the rock. Separately, their results also imply that current knowledge of phosphorus diffusion in olivine needs to be revisited, although this is not stated explicitly. The authors then construct a quantitative thermal model for the size of the 76535 magma body and propose a thought-provoking qualitative model for the formation of Mg-suite (or at least troctolite) parental magmas that I believe constitutes a solid step forward.

These results will undoubtedly be of interest to the lunar science community. The observations are sound and the data collected are robust. The modeling efforts seem well-conceived and for the most part consider appropriate ranges for uncertain parameters. The paper is generally well-written, although it would benefit from a more careful read through. I have provided some comments below for the authors to consider and incorporate, however I see no barriers to publication and strongly recommend the manuscript to move forward in the publication process at Nature Communications, which is an appropriate journal for this work.

General Comments

I find the petrogenetic model envisioned in figure 5 to be interesting and thought provoking. Although my colleagues and I have long advocated for the hybridized source model for Mg-suite parental melts for a number of reasons, the physical mixing of deep dunitic cumulates with KREEP and crustal anorthosite has always seemed problematic to me. I think this model is a step toward shoring up that weakness. However, my concern is, how do the other Mg-suite lithologies fit into such a model? This paper is obviously about a troctolite, but I don't think one can or should treat their origins piecemeal. The norites and gabbronorites, especially those from Apollo 17, have textures, major element compositions, and REE abundances that indicate they may be direct fractional crystallization products of the same or similar parental magmas as troctolites such as 76535. In a magma chamber model this makes sense: the liquid evolves through fractionation and crystallizes troctolites, norites,

and then gabbronorites. Troctolite parental melts are very olivine normative, so it seems that extensive fractional crystallization is required. In a reactive flow model, however, I do not see an obvious way to produce these other Mg-suite lithologies. So while I like the model proposed here and think it represents some creative thought regarding Mg-suite origins, as it is, I think it leads to a whack-a-mole problem: if troctolites formed in this fashion, how then do we make more evolved and just as important lithologies of the Mg-suite? I think the authors should consider this during their revisions and think about physical models that might be able to account for and explain their observations here while also leaving room for the broader Mg-suite.

What we did to address this comment: Added a paragraph discussing how this model could produce other members of the Mg-suite (Lines 221-230).

More Specific Comments

Lines 18-19: Only FANs are crystallization products of the LMO, most lunar lithologies are not.

We thank the reviewer for noticing this potentially misleading sentence.

What we did to address this comment: We modified this sentence to read “This model postulates that the Moon was once largely molten, and the lithologies now present on the Moon are produced from geochemical reservoirs defined by crystallization products of this melt.”

Lines 31 – 37: Carefully revisit the references in these lines. Although the refs cited did indeed invoke the processes they are associated with here, they were not the first papers to suggest those processes, as is stated.

What we did to address this comment: We updated the references, adding the earlier versions of the studies (Hess et al., 1978; as well as Warren and Wasson, 1979) , and rewording to clarify which studies contributed to each finding. We kept the original citations in as well, as they preserve the most relevant discussion for an unfamiliar reader.

Line 87: I believe there’s a typo in the subscript of D-Plag. Should be Ca-Na, not Ca-Mg.

We appreciate the reviewer noticing this typo, and it has been fixed.

Line 95: Delete 'other', P is not tetravalent.

Agreed.

What we did We modified this sentence accordingly (lines 121-123).

Line 103: Please define what functions and coefficients are being referred to. Cooling pathways for functions or did the authors use different equations? Does 'coefficients' refer to diffusion coefficients or the fit coefficients (a and b)? If 'coefficients' refers to varying possible diffusion coefficients, insert 'diffusion' here so as to be specific.

We added 'fit' for clarity (line 130)

Lines 114 – 115: References for previous estimates?

We added the following references: Borg et al., 2017; Gooley et al., 1974; Stewart, 1975; McCallum et al., 2006

Line 116: This probably does not need to be its own section, as the recrystallization section is only a paragraph long. More broadly, I think the authors can rethink their sectioning of the paper to reduce the number, this being one example.

We followed the reviewer's suggestion and combined this section with the "Chemical heterogeneities" section

Lines 198 – 204: Give the amount of annealing and recrystallization that has occurred in 76535, I am unsure how any of the plag grains in the rock meet criteria (a) here. Please provide more info.

To clarify this section, we added "All four grains are surrounded by either olivine or orthopyroxene, preserving igneous growth facets."

Line 342: Typo? Should read 'equation 3' instead of 4, I believe.

Done, replaced '4' with '3'

Lines 377 – 381: I'm not sure I agree that we have no independent constraints on crustal temperatures, or that the crust was as cool as assumed here. Since emplacement of the Mg-suite pretty much has to occur essentially contemporaneously with the end of the LMO, the crust, especially the lower crust, can't be much below its solidus. Additionally, work by Meyer et al. (2010) *Icarus* has modeled the temperature evolution of the crust (and the effects of tidal heating on it), and in my 2020 *Nature Geoscience* paper, Matthieu Laneuville's modeling of heat production from KREEP shows just how much heat is produced from radioactive decay, which will undoubtedly keep the crust quite warm. The two-pyroxene temperatures from McCallum and Schwartz (2001) and my 2012 paper both show 76535 sitting at 800 – 900 C when (presumably) it was excavated from the lower crust, so those temps, combined with the very slow cooling rate from Borg et al. suggest the crust was warmer than assumed here. I think it's worth checking out the references here, especially the Meyer paper, and consider a thermal model with a hotter crust.

As discussed with the first reviewer's last comment, we modeled at warmer temperatures, and made Supplementary Figure 8 to consider the effect of a warmer crust.

References: My 2020 *Nature Geoscience* paper is in the reference list twice (8 and 38). Although I'd be happy to have this count as two citations instead of one, it won't work that way, so better to just cite it once.

Fixed!

Steve Elardo
University of Florida
5/20/2021

Reviewer 3:

Review of manuscript 307314 for Nature Communications:

Fast magmatic cooling of the slowest-cooled Apollo sample

by W. S. Nelson, J. E. Hammer, T. Shea, E. Hellebrand, and G. J. Taylor

This manuscript reports important significant observations on lunar sample 76535, the much-studied and photogenic archetype of the lunar Mg-suite plutonic rocks. The manuscript should be published in nearly its present form. I found it to be well-written, well-organized, and compelling.

My comments below should be dealt with easily – as mostly they are addenda to the arguments. One will require a bit of reconsideration or presentation of data not already in the manuscript.

General Comments

1. The authors' discussion of zoning in the 76535 olivine and modeling thereof (page 3, Fig. 1) needs to include information on the orientation of the P-enriched lamellae relative to the thin section plane. This is important, because this effect of thin section geometry can lead to uncertainties of an order of magnitude in diffusion timescales because of geometry and diffusion anisotropy (Shea et al. 2015). This uncertainty affects diffusion modeling of the widths of the P-rich lamellae, so that lamellae at lower angles with respect to the section plane should appear to widen faster than those at high angles – this difference is not apparent in the modeled profiles (Figs. 1, 3).

It would be helpful if the authors could define the crystallographic planes on which the P-rich lamellae lie – perhaps by analogy with experimentally produced lamellae or well-characterized natural ones. Alternatively, perhaps the authors could determine the orientations of the olivine grains by EBSD. Then, they could calculate the angles between the section plane and the lamellae's planes.

Unfortunately, knowledge of the section orientation with respect to an olivine crystal is not sufficient to resolve the orientation of P-rich lamellae within the crystals. We know of no published description of the 3D distribution of P heterogeneities, but the available maps of oriented sections suggest that (1) primary branches are unlikely to parallel the principal crystallographic directions (e.g. Welsch et al. 2014), and (2) secondary branches are parallel to crystal faces (prominent forms such as $\{110\}$, $\{021\}$) rather than the crystal axes.

Importantly, the sectioning issues raised by this reviewer do not jeopardize our interpretations and retrieved time scales for progress toward diffusive homogenization. The length scale of lamellae we observed with EMP x-ray intensity mapping is already at the lower limit of spatial resolution for the technique. Sectioning effects (cutting the lamellae at high angles) would result in artificial widening of lamellae; downward-adjustment of lamellae thickness to account for oblique sectioning would only shorten the retrieved diffusion timescales.

What we have done to address this comment: We collected EBSD data and added Supplementary Figure 8 to demonstrate the crystallographic orientation of these crystals. We also collected and

present orientation information for the plagioclase crystals included in the diffusion modeling.

2. The authors explain the absence of lamellae in most olivines, and most parts of the olivines with lamellae, by dissolution/reprecipitation during a later event of partial melting initiated by infiltration of basaltic melt. I've no argument with this, but homogenous olivine can form in other ways. Most specifically for the olivine of Figure 1f-1i, formation and crystallization of a melt inclusion can erase P zoning. For example, Goodrich et al. (2013) show strong lamellar P-zoning in olivine in the Nakhla martian meteorite, and that melt inclusions in the olivine are surrounded by areas of constant low P (her Figs. 2, 3). These are (or can be) interpreted as cognate olivine, grown from the melt inclusion at leisure as the rock cooled. Another possible mechanism for erasing P-zoning is migration and coalescence of melt inclusions. In a rapidly grown dendritic olivine, trapped melt will lie in elongated discs or rods among the dendrites, and it can coalesce (by solution & reprecipitation) to single larger inclusions to minimize surface area/energy.

Though this is an interesting concept, it seems extremely unlikely for the crystals studied here. P-poor areas extend from edge to edge in several grains, and they completely surround the P-rich zones. The truncations are also quite different to those around melt inclusions. The lamellae in 76535 are far sharper and do not come to a point as those presented in the Goodrich study. Further, the absence of surviving melt inclusions in 76535 argues against this hypothesis.

3. Although lamellar P-zoning is recognized wherever basalts are found, the authors might add references for such zoning in other lunar samples. For example, Elardo and Shearer (2014) for a lunar meteorite basalt and Demidova et al. (2018) for L20 soil grains. Similarly, the authors could point to lunar materials inferred to have cooled slowly, xenoliths in basalt, that do not show zoning in P (Shearer et al. 2015).

Done, we added these references in the section “Chemical Heterogeneities in 76535” section, lines 94-95.

4. I’m unconvinced by the formation scenario section of the manuscript in two respects – the second scenario seems unlikely, and a third (uncited) scenario seems possible. In the second scenario (lines 167 ff), the authors would have “... mafic cumulate diapirs flow[ing] into the lower crust...”. This seems unlikely to me as mafic cumulate diapirs would have been dense, and

the lunar crust, being composed primarily of plagioclase, would have been less dense. Hence, there would be no driving force to make the diapirs rise into the crust – they would stall at the crust-mantle boundary. The authors should explain their model further.

We reworded this section to clarify that we do not expect density driven upwelling of mafic cumulates through the crust, but rather mechanical mixing of the two lithologies during cumulate overturn of the lunar mantle. (209-210)

A third scenario seems possible, that 76535 formed from an impact melt. One can readily imagine an impact large enough to melt and mix material from the crust and uppermost mantle, yielding a melt of the proper composition to parent 76535. That melt would be superheated, and cool quickly down below the olivine-liquidus, allowing rapid growth of olivine and formation of P-rich lamellae. Cooling would slow down considerably thereafter, as the melt would have been emplaced into (or onto) crust that had been strongly heated by the impact event. I've no numbers to go with this scenario, but it seems plausible *prima facie*.

We adopted this suggestion by adding an explicit discussion of the impact melt sheet hypothesis in the revised text. (lines 192-203)

5. The analytical conditions for plagioclase are a concern. I usually analyze plagioclase at much milder conditions, 10-30 nanoamps beam current, because Na can be burnt out of the plagioclase. I wonder if the plagioclase grains without apparent Na zoning were analyzed at high beam currents, like the 100 nA.

The plagioclase in slide , 46 was analyzed at 100 nA's, and revealed the most clear Na distribution in any of our samples. Thus it seems unlikely that Na heterogeneities were preserved in , 46 but not , 150. Further, all profiles were collected at lower currents, as reported in the methods.

Smaller comments

1. The title is confusing, how can it have cooled fast and slow simultaneously? To help

thereader, I recommend revising the title to something like “Rapid magmatic cooling of lunar troctolite 76535 – Evidence against a plutonic origin for this Mg-suite ‘plutonite.’ ”

We changed the title to “Chemical Heterogeneities Reveal Early Rapid Cooling of Apollo Troctolite 76535”

2. References 8 and 38 are the same.

The two references are now combined.

3. Several picky comments through the text, mostly unimportant to the sense of the paper.

REFERENCES.

- Goodrich, C. A., Treiman, A. H., Filiberto, J., Gross, J., & Jercinovic, M. (2013). K₂O rich trapped melt in olivine in the Nakhla meteorite: Implications for petrogenesis of nakhlites and evolution of the Martian mantle. *Meteoritics & Planetary Science*, 48(12), 2371-2405.
- Demidova, S. I., Ntaflos, T., & Brandstätter, F. (2018). P-bearing olivines from the “Luna-20” soil samples, their sources and possible phosphorus substitution mechanisms in lunar olivine. *Petrology*, 26(3), 314-327.
- Elardo, S. M., & Shearer Jr, C. K. (2014). Magma chamber dynamics recorded by oscillatory zoning in pyroxene and olivine phenocrysts in basaltic lunar meteorite Northwest Africa 032. *American Mineralogist*, 99(2-3), 355-368.
- Shea, T., Costa, F., Krimer, D., & Hammer, J. E. (2015). Accuracy of timescales retrieved from diffusion modeling in olivine: A 3D perspective. *American Mineralogist*, 100(10), 2026- 2042.
- Shearer, C. K., Burger, P. V., Bell, A. S., Guan, Y., & Neal, C. R. (2015). Exploring the Moon's surface for remnants of the lunar mantle 1. Dunite xenoliths in mare basalts. A crustal or mantle origin? *Meteoritics & Planetary Science*, 50(8), 1449-1467.

REVIEWERS' COMMENTS

Reviewer #1 (Remarks to the Author):

Nelson et al., have done an excellent job in addressing my previous comments and concerns. My only real major critique of the previous version of this manuscript was that I felt its discussion, in some ways, undersold its important quantitative results--certainly, a vote of confidence in the work presented and not a weakness. Thus, I find that the revised version of this manuscript stands tall on the present analysis and as before, I believe that this analysis provides a substantial advancement to the field. I find that the added discussion of geochronology in the context of their diffusion modeling and constraints will be of significant interest to the broader community. Further, the added and nuanced discussion of how these results fit into our current state of knowledge is a great contribution to lunar science. Likewise, I look forward to the future work that will be motivated by this study. In my opinion, the manuscript should be accepted for publication by Nature Communications.

I caught only one copy-edit: Line 272; "...model ages for the FAN and Mg-suite23. as well as the considerably younger..." should contain a comma, and not a period.

Sincerely,

Tabb C. Prissel

Response to reviewers:

Reviewer #1 (Remarks to the Author):

Nelson et al., have done an excellent job in addressing my previous comments and concerns. My only real major critique of the previous version of this manuscript was that I felt its discussion, in some ways, undersold its important quantitative results--certainly, a vote of confidence in the work presented and not a weakness. Thus, I find that the revised version of this manuscript stands tall on the present analysis and as before, I believe that this analysis provides a substantial advancement to the field. I find that the added discussion of geochronology in the context of their diffusion modeling and constraints will be of significant interest to the broader community. Further, the added and nuanced discussion of how these results fit into our current state of knowledge is a great contribution to lunar science. Likewise, I look forward to the future work that will be motivated by this study. In my opinion, the manuscript should be accepted for publication by Nature Communications.

I caught only one copy-edit: Line 272; "...model ages for the FAN and Mg-suite²³. as well as the considerably younger..." should contain a comma, and not a period.

We replaced the period with a comma, as suggested!

Sincerely,

Tabb C. Prissel